# Four individually identified paired dopamine neurons signal taste punishment in larval *Drosophila*

Denise Weber[1], Katrin Vogt[1,2,3,4], Anton Miroschnikow[5], Michael J Pankratz[5], Andreas S Thum[1,6]*

[1]Department of Genetics, Leipzig University, Leipzig, Germany; [2]Department of Biology, University of Konstanz, Konstanz, Germany; [3]Centre for the Advanced Study of Collective Behaviour, University of Konstanz, Konstanz, Germany; [4]Center for Brain Science, Harvard University, Cambridge, United Kingdom; [5]Department of Molecular Brain Physiology and Behavior, LIMES Institute, University of Bonn, Bonn, Germany; [6]German Centre for Integrative Biodiversity Research (iDiv) Halle-Jena-Leipzig, Leipzig, Germany

*For correspondence:
andreas.thum@uni-leipzig.de

Competing interest: The authors declare that no competing interests exist.

## eLife Assessment

This comprehensive study presents **important** findings that delineate how specific dopaminergic neurons (DANs) instruct aversive learning in *Drosophila larvae* exposed to high salt through an integration of behavioral experiments, imaging, and connectomic analysis. The work reveals how a numerically minimal circuit achieves remarkable functional complexity, with redundancies and synergies within the DL1 cluster that challenge our understanding of how few neurons generate learning behaviors. By establishing a framework for sensory-driven learning pathways, the study makes a **compelling** and substantial contribution to understanding associative conditioning while demonstrating conservation of learning mechanisms across *Drosophila* developmental stages.

**Abstract** Dopaminergic neurons (DANs) play key roles in processing rewards and punishments across species. They evaluate sensory input, store memories, and update them based on relevance. To understand how individual DANs contribute to these functions, we studied *Drosophila* larvae, which have only about 120 DANs. Only eight of these project to the mushroom body (MB), a center for olfactory learning. These eight are divided into the pPAM and DL1 clusters, with four DANs each. We confirmed that pPAM neurons in the MB medial lobe encode sugar rewards. In the DL1 cluster, four neurons—DAN-c1, DAN-d1, DAN-f1, and DAN-g1—each target different MB regions. Notably, optogenetic activation of DAN-f1 and DAN-g1 can substitute for punishment. Additional methods (inhibition, calcium imaging, connectomics) show each DL1 DAN encodes a unique aspect of punishment, with DAN-g1 being pivotal for salt-based signals. Our findings reveal a clear division of labor among larval DL1 DANs for encoding punishment. The striking resemblance in the organizing principle of larval DANs with that of its adult counterpart and the mammalian basal ganglion suggests that there may be a limited number of efficient neural circuit solutions available to address more complex cognitive challenges in nature.

## Introduction

To cope with a constantly changing environment, animals benefit from learning behavioral rules that need constant adaptation and updating. To test this ability, researchers often use Pavlovian or classical conditioning experiments (*Pavlov, 1927*), which examine an organism's capacity to form associations between sensory cues, called conditioned stimuli (CS), and rewards or punishments, referred to as aversive and appetitive unconditioned stimuli (US) or teaching signals (*Gerber et al., 2009*; *Glanzman, 1995*; *Heisenberg, 2003*; *Menzel, 2022*; *Waddell, 2013*; *Weber et al., 2023a*; *Widmann et al., 2018*). DANs and other modulatory neurons mediate teaching signals that allow animals to classify a given CS into positive or negative valence based on past experience (*Cognigni et al., 2018*; *Menzel, 2001*; *Schultz, 2015*; *Thum and Gerber, 2019*; *Watabe-Uchida et al., 2017*). For example, when a CS is paired with simultaneous activation of a specific set of DANs in the PAM cluster, larval or adult *Drosophila* subsequently perceive the odor as attractive (*Burke et al., 2012*; *Eschbach et al., 2020*; *Liu et al., 2012*; *Rohwedder et al., 2016*; *Schroll et al., 2006*) – a similar effect was seen after optogenetic activation of DANs located in the ventral tegmental area of the mouse brain to induce conditioned place preference (*Tsai et al., 2009*). In contrast, activation of larval DL1 DANs (*Eschbach et al., 2020*; *Schroll et al., 2006*; *Weiglein et al., 2021*), adult PPL1 DANs (*Aso et al., 2012*; *Aso and Rubin, 2016*; *Aso et al., 2010*; *Claridge-Chang et al., 2009*), or DANs in the posterior tail of the striatum of mice were shown to reinforce avoidance learning (*Menegas et al., 2018*). These examples illustrate that the dopaminergic system constitutes a highly conserved and fundamental neural principle of the brain, which has been retained throughout the process of evolution and executes comparable functions across diverse animal species, including even humans (*Frick et al., 2022*).

Our research focuses on the *Drosophila* larva, an organism endowed with a relatively simple central nervous system, composed of approximately 12,000 neurons (*Dumstrei et al., 2003*; *Winding et al., 2023*). Classical conditioning in larvae can be analyzed by standardized behavioral assays (*Almeida-Carvalho et al., 2017*; *Apostolopoulou et al., 2013*; *Scherer et al., 2003*; *Weber et al., 2023b*), in conjunction with a wealth of genetic methods for targeting single molecules and neurons (*Dietzl et al., 2007*; *Li et al., 2014*; *Perkins et al., 2015*; *Port et al., 2020*). In addition, the connectome of the brain was recently reconstructed, which enables the extraction and analysis of related wiring motifs, as well as entire learning and memory circuits (*Winding et al., 2023*). Based on these advances, it was possible to establish a more fundamental cellular and synaptic understanding of the larval memory circuit, the MB network. The MB is a higher-order parallel fiber system present in numerous invertebrate brains, including hemimetabolous as well as holometabolous insects and their larval stages (*Strausfeld et al., 1998*; *Strausfeld et al., 2020*). Analogous to the vertebrate cerebrum, the MB in insects is responsible for forming and maintaining associations (*Heisenberg, 2003*; *Menzel, 2001*; *Menzel, 2022*; *Thum and Gerber, 2019*; *Tomer et al., 2010*; *Waddell, 2013*). In the first developmental stage (L1), the larval MB comprises approximately 110 intrinsic Kenyon cells (KC) per hemisphere, which sparsely encode for conditioned stimulus (CS) information and is dominated by olfactory input (*Eichler et al., 2017*; *Masuda-Nakagawa et al., 2009*).

The transmission of olfactory information from the peripheral sensory organ is achieved via only 21 olfactory receptor neurons (ORNs) per body side, which in turn send signals across 35 uni- and multiglomerular projection neurons, ultimately reaching the MB KCs (*Berck et al., 2016*; *Fishilevich et al., 2005*; *Kreher et al., 2005*; *Masuda-Nakagawa et al., 2009*; *Ramaekers et al., 2005*). KCs receive modulatory input from 17 neurons, consisting of eight dopaminergic neurons, four octopaminergic neurons, and five neurons with unknown signaling molecule identity (*Eichler et al., 2017*; *Rohwedder et al., 2016*; *Saumweber et al., 2018*; *Selcho et al., 2009*). The DANs are divided into two clusters based on the location of their soma and related neuronal lineage, namely the primary protocerebral anterior medial cluster (pPAM) and the dorsolateral 1 cluster (DL1). Each DAN innervates a specific and non-overlapping site in the MB, which, together with one to three output neurons, defines a distinct compartment (*Eichler et al., 2017*). The larval MB comprises a total of 11 compartments, eight of which are defined by DAN input (*Eichler et al., 2017*; *Saumweber et al., 2018*). The four DANs of the pPAM cluster (DAN-h1, DAN-i1, DAN-j1, and DAN-k1) innervate the four related compartments (h, i, j, and k) of the medial lobe and provide teaching signals for sugar rewards (*Eichler et al., 2017*; *Rohwedder et al., 2016*; *Schleyer et al., 2020*). The four DANs of the DL1 cluster (DAN-c1, DAN-d1, DAN-f1, and DAN-g1) innervate the lower peduncle (c), the lateral appendix (d), the intermediate (f), and the lower vertical lobe (g) (*Eichler et al., 2017*; *Saumweber et al., 2018*). Previous studies

have shown that the DL1 cluster is capable of mediating aversive teaching signals (*Eschbach et al., 2020*; *Schroll et al., 2006*; *Selcho et al., 2009*; *Weiglein et al., 2021*). Moreover, larvae can use a variety of natural stimuli as aversive teaching signals such as heat, electric shock, vibration, bitter substances (like quinine or caffeine), and high salt concentrations (low salt concentrations, however, are rewarding) (*Aceves-Piña and Quinn, 1979*; *Apostolopoulou et al., 2016*; *Apostolopoulou et al., 2014*; *Eschbach et al., 2011*; *Gerber and Hendel, 2006*; *Khurana et al., 2009*; *Khurana et al., 2012*; *Niewalda et al., 2008*; *Pauls et al., 2010*; *von Essen et al., 2011*). However, it is currently unknown if and how these natural stimuli are represented by the DL1 cluster.

We only know that optogenetic stimulation of nociceptive sensory neurons and their downstream basin neurons has been observed to activate three out of the four DL1 DANs, namely DAN-d1, DAN-f1, and DAN-g1 (*Eschbach et al., 2020*). Furthermore, the simultaneous presentation of odor with optogenetic activation of these three DANs has been found to elicit aversion in larvae (*Eschbach et al., 2020*; *Weiglein et al., 2021*). These findings suggest that the activation of DL1 DANs may induce a punishing effect during training, based on somatosensory perception. But can these four individual DANs also process other aversive sensory modalities such as chemosensory information? Utilizing electron microscopy-based reconstruction, all first and second order input neurons of the MB, consisting of a complete set of 102 neuron pairs, were examined (*Eschbach et al., 2020*; *Winding et al., 2023*). The analysis identified numerous afferent sensory inputs via 20 pairs of feedforward neurons (FFNs), which can now be integrated with the published wiring diagram of all chemosensory input neurons (*Miroschnikow et al., 2018*). This enables a more comprehensive understanding of the distinct modes of action of the four DL1 DANs in relation to the processing of different sensory modalities.

Consequently, we have investigated the role of the four DL1 DANs in the larval brain regarding their involvement in encoding a salt-dependent aversive teaching signal at the single-cell level. Utilizing behavioral assays, fluorescence microscopy, calcium imaging experiments, and connectome analyses, we have established a more comprehensive description of the coding space of the DANs at the single-cell level. Our investigation unveils a cellular specialization among the four DL1 DANs, where individual DANs code for different but partially overlapping aspects of the aversive teaching signal.

## Results

### The larval dopaminergic system is functionally bipartite to convey positive and negative teaching signals

There are eight DANs in the larval brain that innervate the mushroom body that can be anatomically subdivided into four pPAM cluster and four DL1 cluster neurons (*Figure 1A-C*, *Figure 1—figure supplement 1A-C*, and *Figure 1—source data 1*). Three of the four pPAM cluster DANs can be labeled with the driver line R58E02-Gal4 (*Figure 1—figure supplement 1A–E*; *Rohwedder et al., 2016*). Pairing an odor presentation with simultaneous optogenetic blue light activation of R58E02 positive DANs using a mutant microbial-type rhodopsin ChR2$^{XXL}$, which depolarizes neurons artificially (*Dawydow et al., 2014*), establishes an appetitive olfactory memory (*Figure 1—figure supplement 1F and G*; *Rohwedder et al., 2016*). Please be aware that the activation of blue light itself does not cause any disruptive behavioral side effects, as it does not independently induce memory formation. In line with this finding, it was shown that ablation or inhibition of R58E02 positive DANs strongly reduces appetitive olfactory memory (*Rohwedder et al., 2016*) but has no functional significance for aversive olfactory memory (*Figure 1—figure supplement 1H and I*; *Rohwedder et al., 2016*). Therefore, it can be concluded that pPAM cluster DANs are not involved in the encoding of aversive teaching signals. How are aversive teaching signals encoded?

The driver line TH-Gal4 can be used to label the four DANs of the DL1 cluster (*Figure 1A–E*), as well as most other DANs, except for a small subset of DANs, such as pPAM DANs and neurons in the subesophageal zone (*Figure 1D and E*; *Selcho et al., 2009*). By coupling odor presentation with simultaneous optogenetic activation of TH-Gal4 positive DANs through ChR2$^{XXL}$, an aversive olfactory memory can be established (*Schroll et al., 2006*), which was observed after both one (*Figure 1F*) and three (*Figure 1G*) odor-DAN activation pairings. Consistent with these findings, aversive associative memory formed after one or three training cycles of odor high salt pairings were significantly reduced when DANs activity was inhibited optogenetically via GtACR2 (*Figure 1H and I*), a light-gated channel

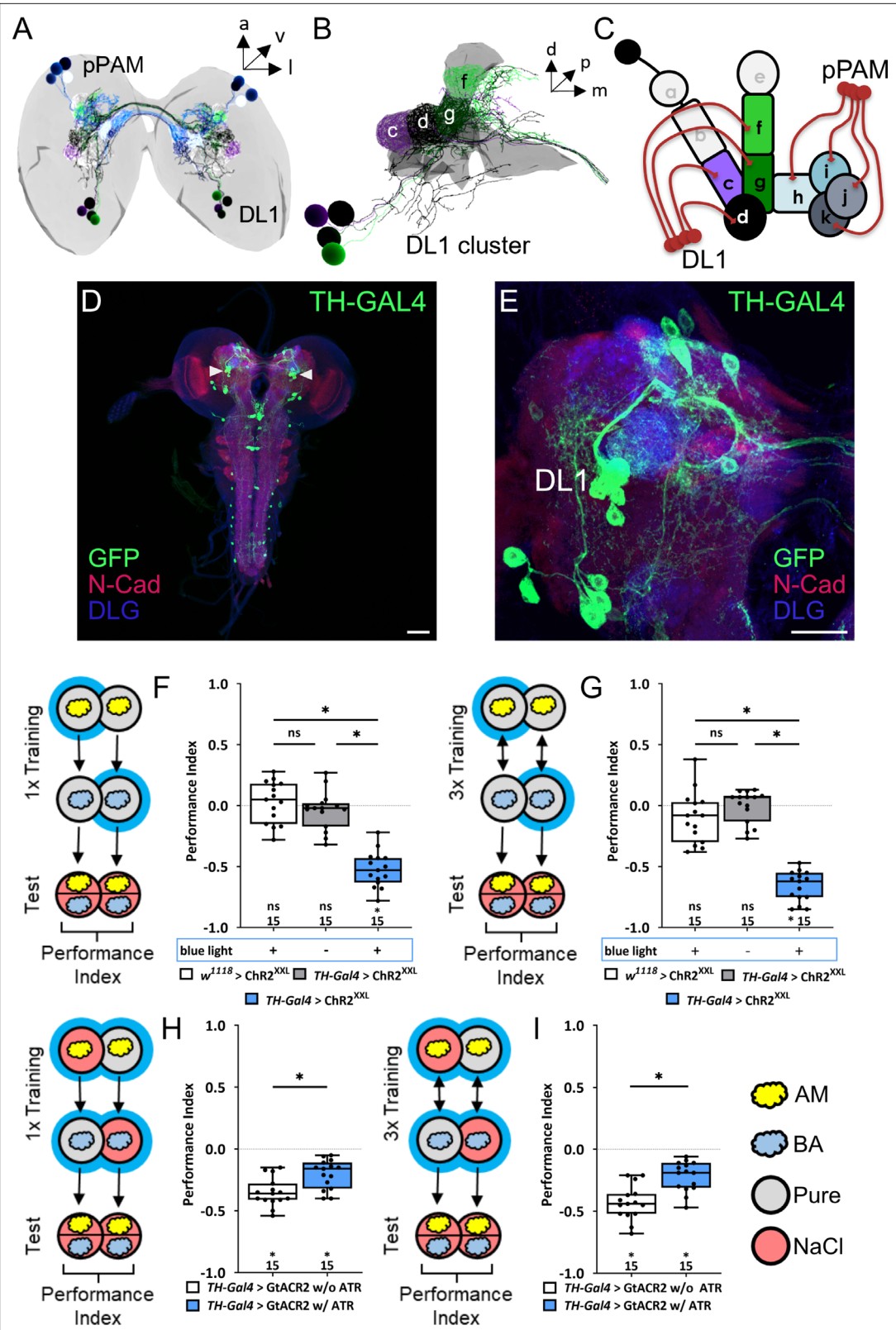

**Figure 1.** The larval dopaminergic system is subdivided into two functionally distinct clusters. (**A**) The larval dopaminergic neurons (DANs) can be anatomically subdivided into the primary protocerebral anterior medial (pPAM) and dorsolateral 1 (DL1) cluster based on their cell body position. (**B**) The DL1 cluster (cell bodies in green and purple) consists of four DANs providing input to the c, d, g, and f compartments of the vertical lobe, peduncle, and lateral appendix of the mushroom body (MB). (in gray). (**C**) Schematic representation of the larval MB using single letter acronyms based on *Saumweber*

*Figure 1 continued on next page*

*Figure 1 continued*

*et al., 2018* to indicate compartment innervation by MB input and output neurons. DL1 DANs specifically innervate the compartments c, d, f, and g in the vertical lobe, lateral appendix, and peduncle, whereas pPAM DANs innervate the shaft (h) and medial lobe (l, j, k) of the MB. (**D, E**) Four DL1 DANs (DAN-c1, DAN-d1, DAN-g1, and DAN-f1) are included in the expression pattern of the TH-Gal4 driver line. However, expression of a UAS-mCD8::GFP reporter via TH-Gal4 labels many more neurons (in green, anti-GFP) throughout the entire CNS (in red and blue, anti-N-cadherin and anti-discs large), in total about 100 neurons. (**F**) To test whether optogenetic activation of the DL1 DANs is sufficient to substitute for a punishment, we used the TH-Gal4 driver in combination with UAS-ChR$^{XXL}$. Experimental and control larvae were trained by simultaneously presenting an odor and blue light and thus artificial activation of DL1 DANs, whereas a second odor was presented in darkness. Only larvae of the experimental genotype (p>0.05, N=15), but not of the two genetic controls (both p<0.05, for each group N=15), retrieve an aversive associative olfactory memory. The same result was seen after one or three training trials (**F** and **G**, respectively). (**H, I**) To test for the acute function of the DL1 DANs in aversive associative olfactory memory, we expressed GtACR2 via the TH-Gal4 driver. Acute optogenetic inhibition of synaptic output from DL1 and other DANs reduced odor high salt memory. Experimental larvae raised on supplemented food (0.5 mM all-*trans*-retinal, ATR) and trained in blue light performed on a lower level than control animals kept on standard food (p<0.05, for each group N=15). A memory impairment was seen after one and three training trials (**G**) and (**H**), respectively. All behavioral data is shown as box plots. Differences between groups are highlighted by horizontal lines between them. Performance indices different from random distribution are indicated below each box plot. The precise sample size is given below each box plot. n.s. p>0.05; *p<0.05. Scale bars: in (**C**) 50 µm and in (**D**) 25 µm. The source data and results of all statistical tests are documented in *Figure 1—source data 1*.

The online version of this article includes the following source data and figure supplement(s) for figure 1:

**Source data 1.** Tables of the raw data with all individual preference indices, the calculated performance indices and the statistical analysis of each experiment.

**Figure supplement 1.** The dopaminergic pPAM cluster encodes for a teaching signal in larvae.

**Figure supplement 1—source data 1.** Tables of the raw data with all individual preference indices, the calculated performance indices and the statistical analysis of each experiment.

from the algae *Guillardia theta* (*Mohammad et al., 2017*). The observed partial reduction may be attributable to biological, technical, or systemic factors, or a combination of these (a more detailed description can be found in the corresponding part of the discussion). In these experiments, salt has to be added during the test to initiate the recall of the aversive memory (indicated as red test plates, *Figure 1F–I*; *Gerber and Hendel, 2006*; *Schleyer et al., 2015*; *Schleyer et al., 2011*). Previous studies have shown that the larva only recalls aversive odor-high salt memory when tested in the presence of the teaching signal. The prevailing explanation suggests that the larva engages in memory-based search during the test, but only if the memory content can be used in a beneficial way (*Gerber and Hendel, 2006*; *Schleyer et al., 2015*; *Schleyer et al., 2011*).

Based on our findings and previously published data on both larval and adult stages, we propose a functional division of labor among DANs projecting to the MB. Specifically, pPAM neurons are likely to convey a dopaminergic appetitive teaching signal, whereas DL1 neurons may transmit a dopaminergic aversive teaching signal. Nevertheless, the findings related to DL1 DANs necessitate additional validation as the TH-Gal4 driver line used in the experiment labels approximately 120 cells, thus limiting the ability to draw cell-specific conclusions.

## Anatomical single-cell analysis of DL1 DANs

In an earlier investigation, we conducted a screening of several thousand Gal4 lines in order to construct a high-resolution atlas of the mushroom body at the cellular level via light microscopy (*Saumweber et al., 2018*). A subsequent reconstruction of the wiring diagram of the entire MB at the electron microscopic level encompassed not only individual neurons but all synaptic connections (*Eichler et al., 2017*). Utilizing this anatomical work, we identified a total of 102 Gal4 lines featuring MB extrinsic neuron expression patterns. These lines were subjected to intersectional strategies in order to restrict expression patterns to single neurons (*Eschbach et al., 2020*; *Saumweber et al., 2018*). For the present study, we employed eight split-Gal4 lines chosen from this set, as they contained only one or two DANs of the DL1 cluster (*Figure 2*).

DAN-c1 was specifically labeled by SS02160 (*Figure 2B*), DAN-d1 by MB328B and MB143B (*Figure 2C*, *Figure 2—figure supplement 1B*), DAN-f1 by SS02180 and MB145B (*Figure 2D*, *Figure 2—figure supplement 1C*) and DAN-g1 by SS01716 (*Figure 2E*). In addition, we used the split-Gal4 lines MB065B (*Figure 2F*), SS01702 (*Figure 2G*) and MB054B (*Figure 2H-K*) that specifically labeled DAN-c1/DAN-f1, DAN-c1/MBIN-e1, or DAN-f1/DAN-g1, respectively. For our evaluation, we crossed each split-Gal4 line with the reporter strain UAS-mCD8::GFP;mb247-LexA,lexAop-mRFP

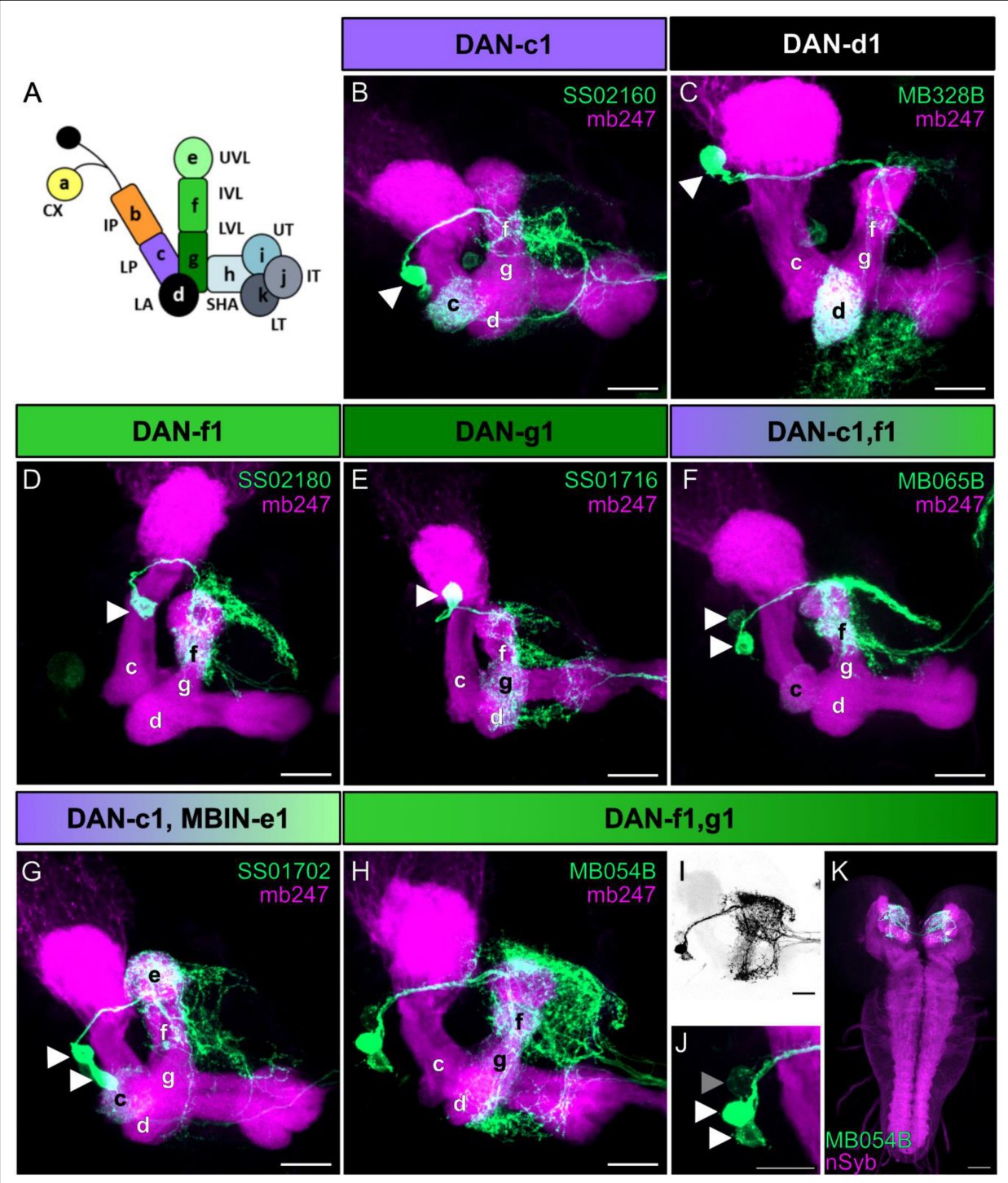

**Figure 2.** Anatomical single-cell analysis of DL1 dopaminergic neuron (DAN) specific split-Gal4 driver lines. (**A**) The larval mushroom body (MB) is organized into 11 compartments: CX calyx; IP and LP intermediate and lower peduncle; LA lateral appendix; UVL, IVL, and LVL upper, intermediate, and lower vertical lobe; SHA, UT, IT, LT shaft as well as upper, intermediate, and lower toe of the medial lobe. Single-letter synonyms of compartment names are given as 'a–k.' These letters are used to indicate compartment innervation by the MB input and output neurons (*Saumweber et al., 2018*). DL1 cluster DANs are DAN-c1, DAN-d1, DAN-g1, and DAN-f1 that innervate the respective four different compartments of the MB. (**B–J**) Individual split-Gal4 driver lines were crossed with the reporter strain UAS-mCD8::GFP;mb247-lexA,lexAop-mRFP. Third instar larval brains were dissected, fixed, and mounted to visualize the fluorescent reporter signal labeling the MB (mb247-lexA, lexAop-mRFP shown in magenta) and the respective DAN pattern (GFP shown in green). (**B–E**) SS02160 (DAN-c1), MB328B (DAN-d1), SS02180 (DAN-f1), SS01716 (DAN-g1) each specifically label a single DL1 DAN (cell bodies are highlighted by white arrowheads). (**F–K**) Two neurons can be seen in MB065B, SS01702, and MB054B split-Gal4 that express in

*Figure 2 continued on next page*

*Figure 2 continued*

DAN-c1/DAN-f1, DAN-c1/MBIN-e1, and DAN-f1/DAN-g1. Please note that MB065B shows strong expression in DAN-f1 but weaker staining in DAN-c1. (**H–K**) MB054B showed reliable strong expression in DAN-f1 and DAN-g1. In some brains, a third weak cell body was visible right next to the other two DANs (**I, J**); GFP channel inverted and shown in black; (**I**; cell body highlighted with gray arrowhead). Due to the low expression level, we were not able to identify this cell given that only the g and f compartments of the MB were innervated (**H**). (**K**) Analysis of the entire brain via native fluorescence expression of GFP (green) and n-Syb (magenta) did not reveal additional cells for MB054B split-Gal4. Scale bars: (**B–J**) 20 μm, (**K**) 50 μm.

The online version of this article includes the following source data and figure supplement(s) for figure 2:

**Figure supplement 1.** Ablation of individual DL dopaminergic neurons (DANs) does not impair aversive olfactory memory reinforced by high salt.

**Figure supplement 1—source data 1.** Tables of the raw data with all individual preference indices, the calculated performance indices and the statistical analysis of each experiment.

**Figure supplement 2.** Split-Gal4 MB054B covers three dopaminergic neurons in the DL1 cluster.

(*Burke et al., 2012*) that allowed us to monitor the expression pattern (green) in the background of a MB specific reference staining (magenta). Our results confirm the reported specificity even despite the use of different reporter lines and lacking antibody staining (*Eschbach et al., 2020*), as we analyzed only endogenous GFP expression using a more cautious and faster whole mount staining protocol (see Material and Methods). In addition, we also verified that the DANs labeled via MB054B are indeed dopaminergic as the GFP expression overlaps with an anti-TH antibody staining (*Figure 2— figure supplement 2*). Having established anatomical validity, we subsequently employed these lines to explore the physiological response of each DAN within the DL1 cluster to sugar and salt stimuli. For the purpose of analyzing DAN-d1 and DAN-f1, we limited our investigation to the split-Gal4 lines MB328B and SS02180 as these lines displayed more robust expression levels.

## Physiological single-cell analysis of DL1 DANs

In order to investigate how high salt teaching signals are represented at the level of individual DL1 DANs, we conducted calcium imaging experiments using a microfluidic chip on intact and immobilized larvae, as previously described (*Si et al., 2019*). We expressed GCaMP6m under the control of four different split-Gal4 lines specific to DL1 DANs (SS02160 (DAN-c1), MB328B (DAN-d1), SS02180 (DAN-f1), and SS01716 (DAN-g1)), as well as the pPAM specific R58E02-Gal4, and exposed the larvae to solutions containing 100 mM salt, 1 M salt, or 500 mM fructose. Our results showed that only DAN-c1 responded to 100 mM salt (*Figure 3A*). While DAN-c1, DAN-d1, and DAN-g1 responded to 1 M salt (*Figure 3A, B and D*). Conversely, pPAM DANs showed a reduced signal in response to 1 M salt (*Figure 3E*). Both DAN-c1 and pPAM DANs showed a calcium increase in response to 500 mM fructose (*Figure 3A and E*). Although DAN-f1 did not respond to either salt concentration, a reduction in the calcium signal was seen for 500 mM fructose (*Figure 3C*). Overall, these results provide support for the functional division of the larval dopaminergic system at the physiological level, where most DL1 neurons are activated by high salt concentrations, with one exception, which is inhibited by fructose. In contrast, pPAM cluster neurons display the opposite response, being inhibited by high salt concentrations and strongly activated by fructose.

## Behavioral single-cell analysis of DL1 DANs

Next, we analyzed the role of single DL1 DANs in aversive olfactory learning (*Figure 2—figure supplement 1D–G*, *Figure 2—figure supplement 1—source data 1*, *Figure 4—figure supplements 1–4*, *Figure 4—figure supplement 1—source data 1*, *Figure 4—figure supplement 2—source data 1*, *Figure 4—figure supplement 3—source data 1*, *Figure 4—figure supplement 4—source data 1*, *Figure 4—figure supplement 5—source data 1*). We employed split-Gal4 lines SS02160 (DAN-c1), MB328B (DAN-d1), SS02180 (DAN-f1), and SS01716 (DAN-g1) to express the apoptosis proteins Hid and Reaper, resulting in the ablation of the respective individual DL1 DANs. The ablated animals were subjected to the aversive odor-high salt memory assay, using one (*Figure 4—figure supplement 1A–D*) and three training cycles (*Figure 4—figure supplement 1E–H*). We utilized two different training protocols to differentiate between the effects on short-term memory (STM) and anesthesia-resistant memory (ARM), as previous studies have demonstrated that larvae exhibit both after one training cycle, whereas only ARM is present after three training cycles (*Widmann et al., 2016*). However, we found that the ablation of individual DL1 DANs had no effect on aversive

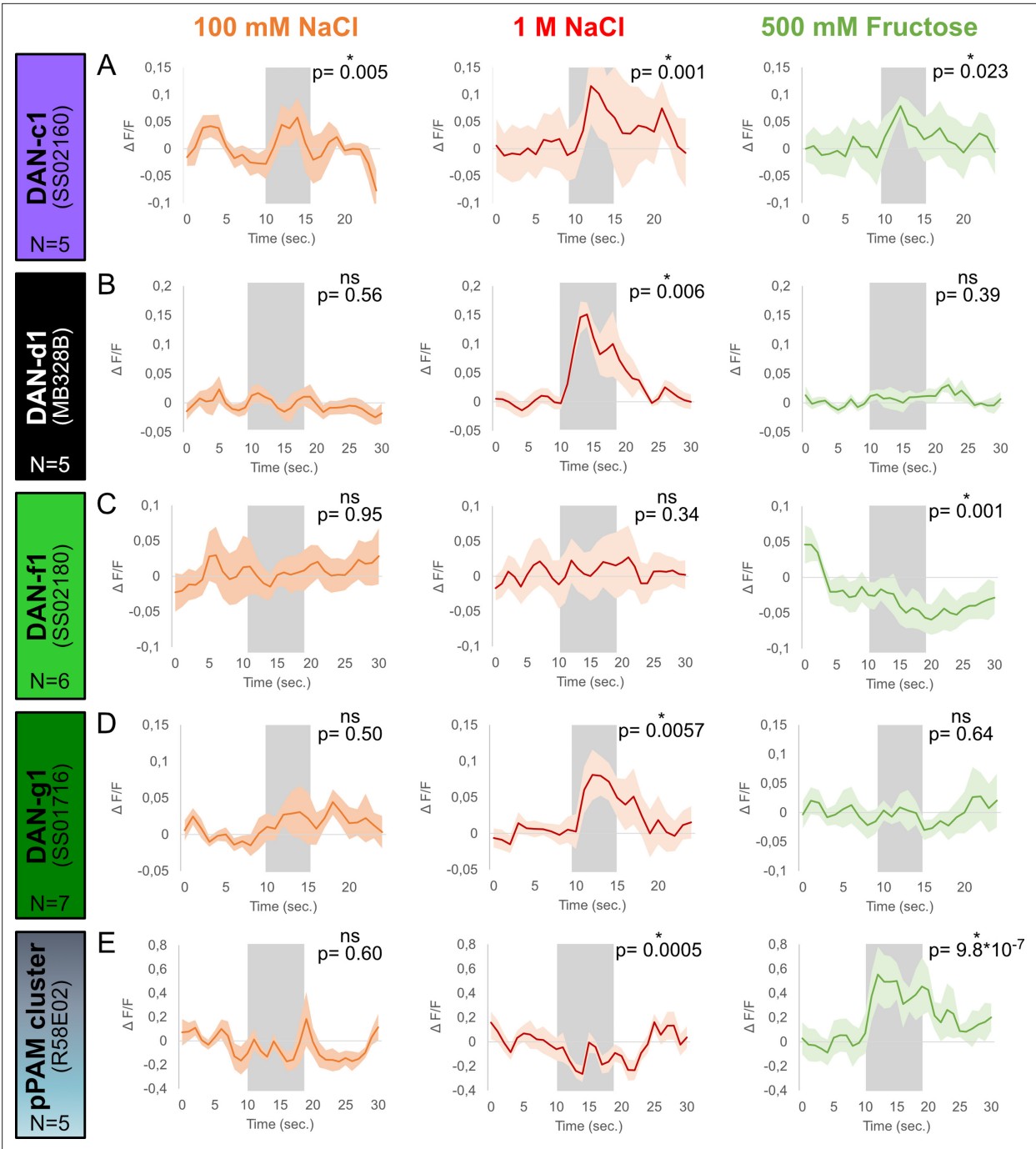

**Figure 3.** Calcium responses of dopaminergic neurons (DANs) to gustatory stimulation. Four different split-Gal4 lines and the R58E02 driver line were crossed with UAS-GCaMP6m to express a calcium reporter in DANs. The responses of each DAN towards 100 mM NaCl, 1 M NaCl, and 500 mM fructose was tested in intact larvae using a microfluidic chip-based setup. (**A**) Calcium responses in DAN-c1 (N=5) were induced by gustatory stimulation with 100 mM NaCl (orange), 1 M NaCl (red), and 500 mM fructose (green, for all p<0.05). (**B**) DAN-d1 (N=5) calcium responses were only seen after 1 M NaCl stimulation (red, p<0.05), but not after 100 mM NaCl (orange, p>0.05) and 500 mM fructose stimulation (green, p>0.05). (**C**) Stimulation with 100 mM (orange, p>0.05) and 1 M NaCl (red, p>0.05) did not induce calcium responses in DAN-f1 (N=6). However, stimulation with 500 mM fructose reduced the obtained calcium signal (green, p<0.05). (**D**) 1 M NaCl (red, p<0.05), but not 100 mM NaCl (orange, p>0.05) and 500 mM fructose (green, p>0.05), induced a calcium response in DAN-g1 (N=7). (**E**) pPAM DANs calcium responses (N=5) were only seen after 500 mM fructose stimulation (green, p<0.05). Stimulation with low (orange) and high salt concentrations (red) did not increase calcium signals; however, both reduced pPAM activity (p<0.05). Each graph shows the mean calcium signal plotted as the relative response strength ΔF/F and the related standard error of the mean on the y-axis. The time in seconds is given below each graph on the x-axis. The gray box indicates the duration of the stimulus application. The sample size

*Figure 3 continued on next page*

*Figure 3 continued*

of each group (N=5–7) is given above each row. n.s. p>0.05; *p<0.05. The source data and results of all statistical tests are documented in ***Figure 3—source data 1***.

The online version of this article includes the following source data for figure 3:

**Source data 1.** Tables of the raw data with all individual preference indices, the calculated performance indices and the statistical analysis of each experiment.

olfactory memories in any of the eight experiments conducted (***Figure 4—figure supplement 1A–H***). For DAN-d1 and DAN-f1, we verified these results by using the additional split-Gal4 lines MB143B and MB145B (***Figure 2—figure supplement 1D–G***). In addition, ablation of DAN-d1, DAN-f1, and DAN-g1 individually did not have any impact on odor-quinine and odor-fructose memories established after one cycle training (***Figure 4—figure supplements 2 and 3***). Moreover, MB input neurons that are not dopaminergic (OAN-a1, a2, MBIN-b1, b2, and OAN-e1) were also found to have no relevant effect (***Figure 4—figure supplement 4***). These results suggest that although DL1 DANs respond to high salt concentrations (***Figure 3***), they are not individually necessary in ablation experiments for mediating the aversive teaching signal. The data suggest that the aversive teaching signal may be distributed across multiple neurons, indicating possible functional redundancy or, alternatively, developmental compensation that could attenuate the effects of DL1 cell ablation. This hypothesis was analyzed later in optogenetic inhibition experiments.

## The combination of DAN-f1 and DAN-g1 is required to establish aversive odor-high salt memories

In order to investigate a possible redundancy among DANs, we proceeded by using split-Gal4 lines MB054B (DAN-f1/DAN-g1), MB065B (DAN-c1/DAN-f1), and SS01702 (DAN-c1/MBIN-e1) to express apoptosis proteins Hid and Reaper. This approach selectively ablated either two DL1 DANs or one DL1 DAN and another MB input neuron with unknown neurotransmitter identity. It should be noted that no additional split-Gal4 lines are presently available for further DL1 DAN combinations or the complete DL1 cluster. We then subjected the larvae to the aversive odor-high salt memory assay, using one (***Figure 4A–C***) and three (***Figure 4D–F***) training cycles. Only larvae with ablated DAN-f1/DAN-g1 exhibited significantly reduced aversive olfactory memory after one cycle training (***Figure 4A***). No memory reduction was observed when ablating DAN-c1/DAN-f1 or DAN-c1/MBIN-e1 (***Figure 4B and C***). Ablation of all three sets of neurons, including MB054B, had no effect on aversive olfactory memory after three training cycles (***Figure 4D–F***). To ascertain the independence of the DAN-f1/DAN-g1 neuron-mediated impairment of aversive olfactory memory after one cycle training from the specific odors used, hexyl acetate and benzaldehyde were employed instead of amyl acetate and benzaldehyde. The results of these experiments confirmed the previous findings, as ablation of DAN-f1/DAN-g1 neurons significantly reduced aversive odor-high salt memory after one cycle training (***Figure 4G***), but not after three cycles (***Figure 4J***). Innate olfactory and gustatory behavior was not impaired by DAN-f1/DAN-g1 ablation (***Figure 4—figure supplement 5***, ***Figure 4—figure supplement 5—source data 1***).

Subsequently, we investigated the selectivity of DAN-f1/DAN-g1 neurons concerning the nature of the aversive teaching stimulus. To this end, we utilized quinine as an aversive teaching signal, a stimulus that has been demonstrated to be learnable by the larva, as evidenced by previous studies (***Apostolopoulou et al., 2014***; ***El-Keredy et al., 2012***). Removal of the DAN-f1/DAN-g1 neurons using the split-Gal4 line MB054B had no effect on aversive odor-quinine memory independent of the number of training cycles (***Figure 4H and K***). Furthermore, ablation of DAN-f1/DAN-g1 did not alter appetitive odor-fructose memories after one or three training cycles (***Figure 4I and L***). Therefore, the DL1 DAN-f1/DAN-g1 combination seems to be specifically required for aversive odor-high salt memory. This is further supported by anatomical data that verified the effectiveness of Hid and Reaper using the MB054B split-Gal4 line (***Figure 4—figure supplement 6***).

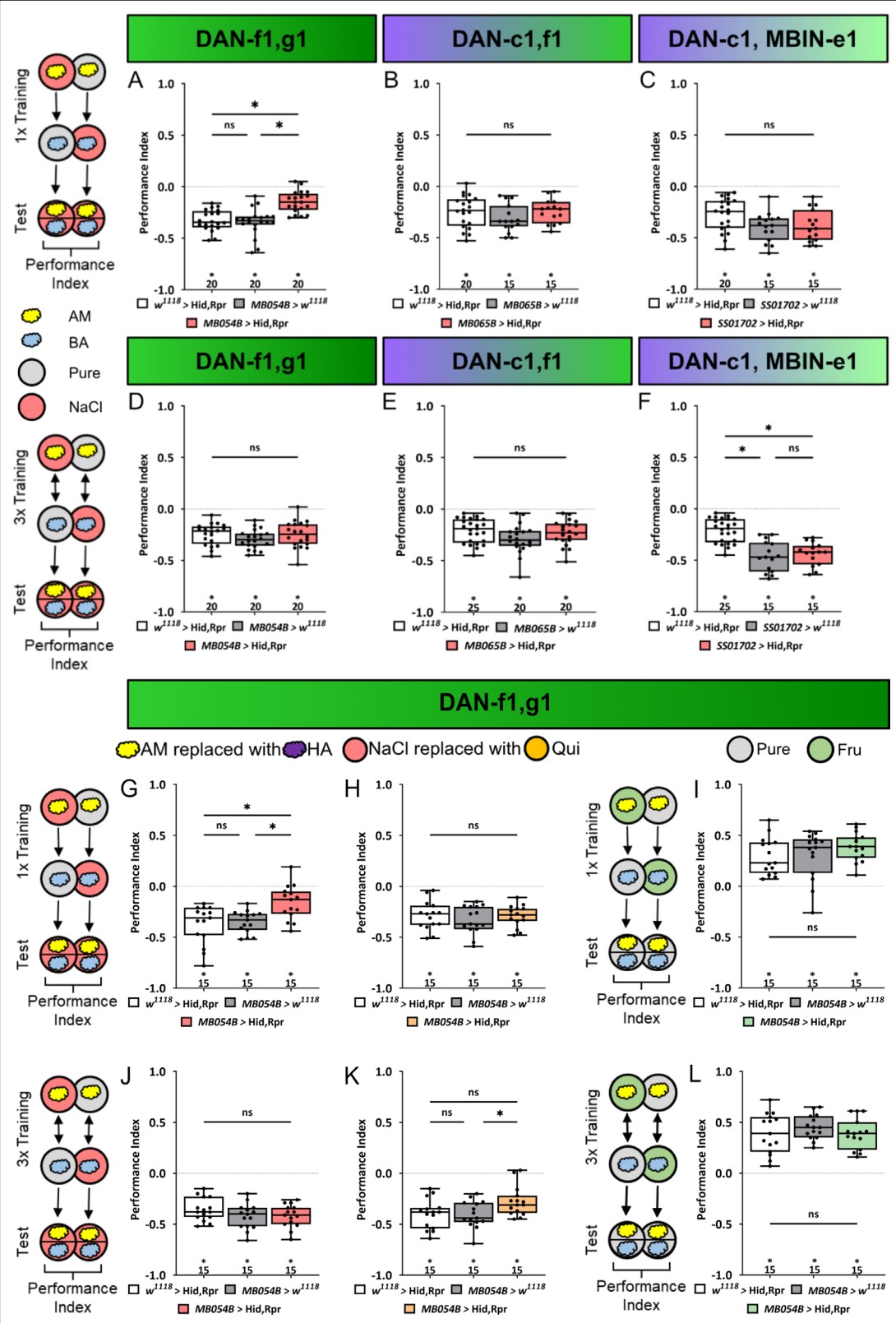

**Figure 4.** Ablation of dopaminergic neuron DAN-f1 and DAN-g1 together impairs aversive olfactory memory. In all panels, associative performance indices are shown for tests immediately after classical conditioning. In the upper panels (**A–C**), larvae are trained once by pairing an olfactory stimulus with high salt punishment, whereas in the lower panels (**D–F**), three training cycles were applied. Schematic overviews for both conditioning protocols are shown on the left. The three different DL1 DAN specific split-Gal4 driver MB054B, MB065B, and SS01702 that each label two neurons were crossed

*Figure 4 continued*

to the effector UAS-hid,rpr to induce apoptosis (**A–F**). (**A**) With MB054B used as driver strain to ablate the DL1 DAN combination DAN-f1/DAN-g1, the aversive associative performance index of the experimental group was decreased compared to both controls (p<0.05, N=15 for each group). (**B–F**) In all other experiments ablation of different DL1 DAN combinations did not reveal a phenotype. (In all experiments, at least one or even both control groups are compared to the experimental group p>0.05, for each group N=15–25). Please note that this also includes MB054B crossed with UAS-hid, rpr tested after three training trials (**D**). (**G**) To verify the memory phenotype of MB054B crossed with UAS-hid,rpr tested after one trial conditioning, we repeated the experiment using the odor pair hexyl acetate (HA) and benzaldehyde (BA). Again, experimental larvae tested after one trial learning showed a robust decrease in odor-high salt memory when compared to both genetic control groups (p<0.05, for each group N=15). The memory phenotype was not seen after three training trials (**J**, p>0.05, for each group N=15). (**H, K**) With MB054B used as driver strain to ablate the DL1 DAN combination DAN-f1/DAN-g1, aversive odor-quinine memory was not impaired after one or three cycle conditioning (p>0.05 when comparing experimental and control groups, for each group N=15). (**I, L**) Similarly, appetitive odor-fructose learning after one and three cycle conditioning was not impaired when ablating DAN-f1/DAN-g1 (p>0.05 when comparing experimental and control groups, for each group N=15). All behavioral data is shown as box plots. Please note that the same data set for the effector control was plotted in subfigures E and F as respective behavioral experiments have been conducted in parallel. Differences between groups are highlighted by horizontal lines between them. Performance indices different from random distribution are indicated below each box plot. The precise sample size of each group is given below each box plot. n.s. p>0.05; *p<0.05. The source data and results of all statistical tests are documented in *Figure 4—source data 1*.

The online version of this article includes the following source data and figure supplement(s) for figure 4:

**Source data 1.** Tables of the raw data with all individual preference indices, the calculated performance indices and the statistical analysis of each experiment.

**Figure supplement 1.** Ablation of individual dopaminergic neurons (DANs) does not impair aversive olfactory memory.

**Figure supplement 1—source data 1.** Tables of the raw data with all individual preference indices, the calculated performance indices and the statistical analysis of each experiment.

**Figure supplement 2.** Ablation of individual DL dopaminergic neurons (DANs) does not impair aversive olfactory memory reinforced by bitter quinine.

**Figure supplement 2—source data 1.** Tables of the raw data with all individual preference indices, the calculated performance indices and the statistical analysis of each experiment.

**Figure supplement 3.** Ablation of individual DL dopaminergic neurons (DANs) does not impair appetitive olfactory memory.

**Figure supplement 3—source data 1.** Tables of the raw data with all individual preference indices, the calculated performance indices and the statistical analysis of each experiment.

**Figure supplement 4.** Functional analysis of mushroom body input neurons that are not dopaminergic.

**Figure supplement 4—source data 1.** Tables of the raw data with all individual preference indices, the calculated performance indices and the statistical analysis of each experiment.

**Figure supplement 5.** Sensory acuity tests.

**Figure supplement 5—source data 1.** Tables of the raw data with all individual preference indices, the calculated performance indices and the statistical analysis of each experiment.

**Figure supplement 6.** Anatomical validation of effector UAS-hid,rpr functionality to ablate neurons of MB054B Split-Gal4 in the DL1 cluster.

## Individual DL1 DANs can instruct an aversive memory recalled on a salt plate

In order to determine if the activation of individual DL1 DANs signals aspects of the natural high salt punishment, we conducted a one training trial learning experiment using ChR2$^{XXL}$ to artificially activate DAN-c1, DAN-d1, DAN-f1, DAN-g1, and the combination of DAN-f1/DAN-g1. We used split-Gal4 lines SS02160, MB328B, SS02180, SS01716, and MB054B, respectively (*Figure 5*, *Figure 5—source data 1*). Simultaneously activating DAN-f1/DAN-g1 via ChR2$^{XXL}$ during an odor presentation establishes an aversive olfactory memory (*Figure 5E and F*). These results are comparable to those observed for TH-Gal4 (as shown in *Figure 1F and G*). Additionally, this memory is observed after one (*Figure 5E*) and three training cycles (*Figure 5F*). Furthermore, single-cell analysis indicates that individual activation of DAN-f1 and DAN-g1, but not of DAN-d1 and DAN-c1 (please note that experimental animals are different from zero and significantly different to one control), can induce an aversive odor memory (*Figure 5A–D*).

## An individual DL1 DAN is acutely necessary to establish an aversive odor-high salt memory

Next, we examined the extent to which the individual DL1 DANs are acutely required for the transmission of a functional aversive teaching signal. We have already shown that the combination of DAN-f1/

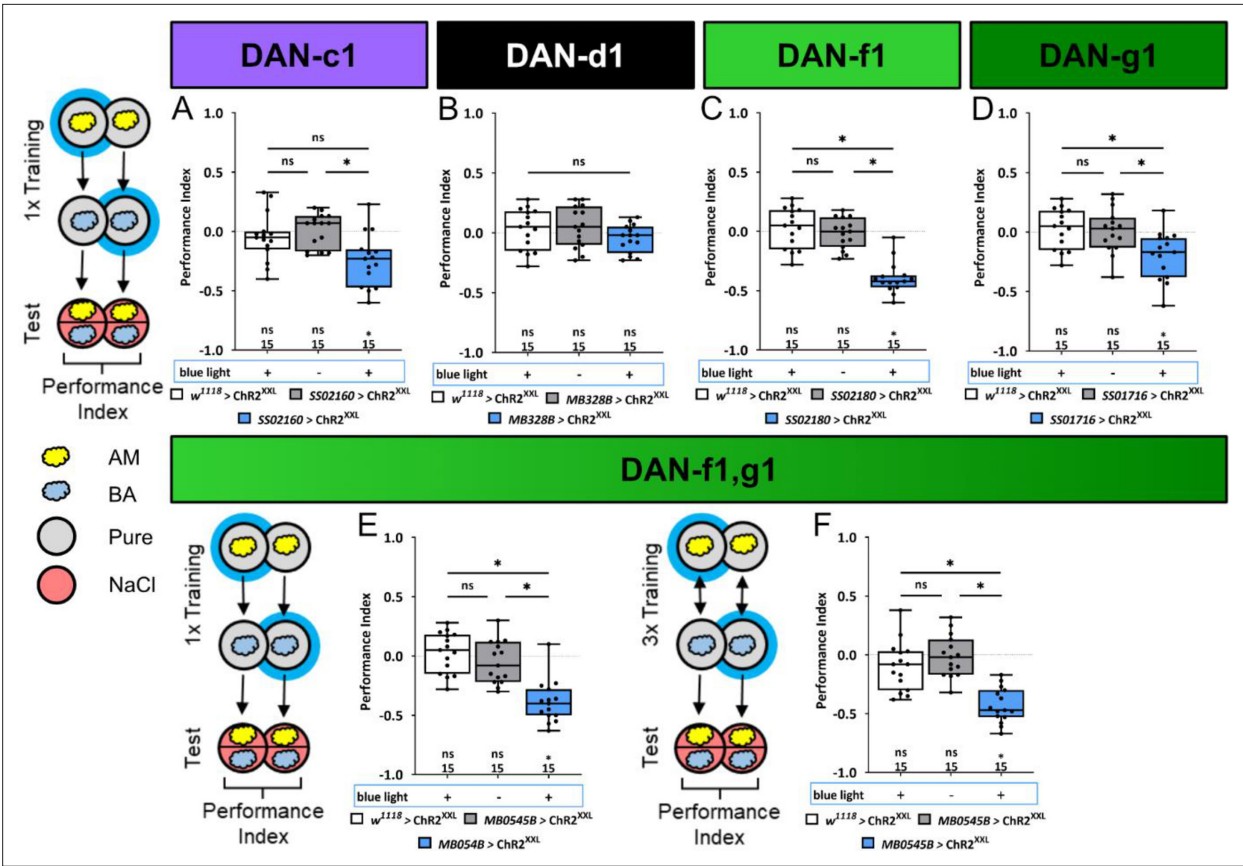

**Figure 5.** Optogenetic DL1 dopaminergic neuron (DAN) activity can substitute for salt punishment. In all panels, associative performance indices are shown for tests immediately after classical conditioning. In panels (**A–E**), larvae are trained once by pairing an olfactory stimulus with artificial blue light activation, whereas in panel (**F**), three training cycles were applied. Schematic overviews for both conditioning protocols are shown to the left of (**A**) and (**F**). (**A–D**) To test whether optogenetic activation of the individual DL DANs DAN-c1, DAN-d1, DAN-f1, and DAN-g1 is sufficient to substitute for a punishment, we used the split-Gal4 lines SS02160, MB328B, SS02180, and SS01716 in combination with UAS-ChR2$^{XXL}$. (**E, F**) For simultaneous optogenetic activation of DAN-f1/DAN-g1, we used MB054B. Larvae of the experimental genotypes for DAN-c1, DAN-f1, DAN-g1, and the DAN-f1/DAN-g1 combination (for all p<0.05, for each group N=15), but not for DAN-d1 (N=15) and all genetic controls (for all p>0.05, for each group N=15), showed an aversive associative memory. The results imply that in the tested conditions, the punishment signal can be mediated by the artificial activation of all individual DL1 DANs, with the exception of DAN-d1. All behavioral data are shown as box plots. Please note that the same data set for the effector control was plotted in subfigures (**B-E**) as respective behavioral experiments have been conducted in parallel. Differences between groups are highlighted by horizontal lines between them. Performance indices different from random distribution are indicated below each box plot. The sample size of each group (N=15) is given below each box plot. n.s. p>0.05; *p<0.05. The source data and results of all statistical tests are documented in *Figure 5—source data 1*.

The online version of this article includes the following source data for figure 5:

**Source data 1.** Tables of the raw data with all individual preference indices, the calculated performance indices and the statistical analysis of each experiment.

DAN-g1 is required for aversive odor-high salt memory (*Figure 4A and G*). However, in these experiments, DANs were removed during the entire development, which may have led to compensatory mechanisms or side effects.

To overcome this limitation, we left the individual DL1 DANs intact during embryonic and larval development and used GtACR2 to selectively inhibit DAN activity only during training, thus isolating the teaching signals from other DAN functions. Optogenetic inhibition of individual DL1 DANs showed no effect on aversive odor-high salt memories (*Figure 6*, *Figure 6—source data 1*), except for DAN-g1 that led to a small but significant reduction (*Figure 6D*). Optogenetic inhibition of DAN-f1/DAN-g1 resulted in a similar reduced aversive odor-high salt memory. This was the case after one training cycle (*Figure 6E*) and three training cycles (*Figure 6F*, *Figure 6—source data 1*). Thus, the

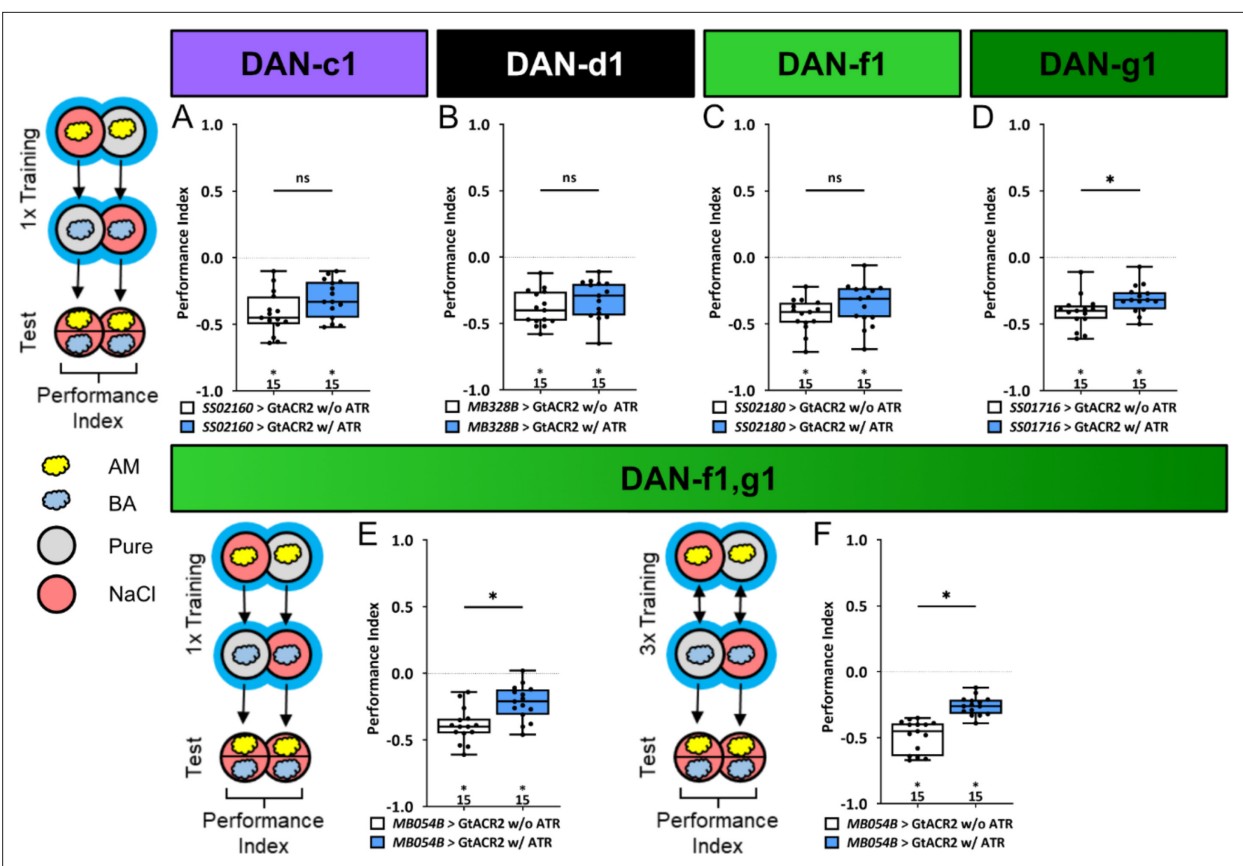

**Figure 6.** Optogenetic inhibition of DL1 DAN activity impairs aversive olfactory memory. In all panels, associative performance indices are shown for tests immediately after classical odor high salt conditioning. In panels (**A–E**) larvae are trained once by pairing an olfactory stimulus with an aversive high salt stimulation, whereas in panel (**F**) three training cycles were applied. Schematic overviews for both conditioning protocols are shown to the left of (**A**) and (**F**). (**A–D**) To test whether optogenetic inhibition of the individual DL DANs DAN-c1, DAN-d1, DAN-f1, and DAN-g1 during training impairs punishment signaling, we used the split-Gal4 lines SS02160, MB328B, SS02180, and SS01716 in combination with UAS-GtACR2 and blue light stimulation during the entire training phase. (**E, F**) For simultaneous optogenetic inhibition of DAN-f1/DAN-g1 we used MB054B. (**A–C**) Larvae with inhibited DAN-c1, DAN-d1, or DAN-f1 function during training showed no impairment of odor-high salt memory comparable to controls that were kept on standard food without supplemented all-*trans*-retinal (ATR, 0.5 mM) and received the same protocol (for all p>0.05, N=15 for each group). (**D–F**) In contrast, inhibition of DAN-g1 alone, or the combination of DAN-f1/DAN-g1 after single trial and three trial conditioning impaired odor-high salt memory compared to controls (for all p<0.05, for each group N=15). This shows that DAN-g1 function is of central importance for signaling a salt punishment teaching signal. All behavioral data is shown as box plots. Differences between groups are highlighted by horizontal lines between them. Performance indices different from random distribution are indicated below each box plot. The precise sample size of each group is given below each box plot. n.s. p>0.05; *p<0.05. The source data and results of all statistical tests are documented in *Figure 6—source data 1*.

The online version of this article includes the following source data for figure 6:

**Source data 1.** Tables of the raw data with all individual preference indices, the calculated performance indices and the statistical analysis of each experiment.

learning deficit in this experiment was more pronounced than after ablation of these two DL1 DANs throughout development (*Figure 4D*).

## A synaptic reconstruction of all chemosensory inputs for each individual DL1 and pPAM DAN

Certainly, a comprehensive analysis from the sensory neurons responsible for salt perception, via the interneurons involved in salt information processing, up to the dopaminergic DL1 teaching neurons would be highly desirable to understand why and how the different DANs respond to salt. Regrettably, such an analysis is currently not feasible due to our limited understanding of salt perception and which sensory cells encode this crucial information (*Liu et al., 2003*). Therefore, this question cannot be answered by the larval connectome. However, it is possible to analyze how qualitative differences

between individual DL1 DANs (*Figures 3–6*) are represented in the connectome, since not all DANs code equally for the punishment teaching signal. Therefore, we compared the input wiring diagrams of the individual DANs of the pPAM and DL1 clusters by mapping the synaptic connections from sensory neurons - to interneurons - to individual DANs, using the full larval brain EM volume (*Figure 7*, *Figure 7—figure supplement 1*; *Eschbach et al., 2020*; *Miroschnikow et al., 2018*; *Winding et al., 2023*).

We found that DAN-c1, DAN-d1, DAN-f1, and DAN-g1 (black circle) clearly differ in their downstream connections. They do, however, receive input from 1, 6, 6, or 11 sensory-to-DAN interneurons (orange circle), respectively (*Figure 7A*), which in turn means that these DL1 DANs are connected to 3, 42, 35, and 9 sensory neurons (numbers in colored circuits below). Please note that we restricted our analysis on direct sensory-to-DANs interneurons (single hop analysis) and did not expand the analysis onto two hop or three-hop connections to keep the evaluation restricted to the shortest paths from sensory neurons to DANs. There is no direct input from sensory neurons onto DL1 DANs. Sensory input thus appears to be increasingly instructing DAN-c1, DAN-d1, DAN-f1, and DAN-g1. The analysis also shows that all four DL1 DANs receive information from all gustatory senses via these interneurons (coded by the yellow color for sensory composition). In addition, DAN-d1, as well as DAN-f1, receive information from olfactory and thermosensory neurons (*Figure 7D*). In general, less processed sensory input seems to be more important for DAN-f1 and DAN-g1 than for DAN-c1 and DAN-d1, as they get 17% and 11% of their entire input from these sensory-to-DAN interneurons in contrast to 1% and 5%, respectively. The four DL1 DANs show a conserved connection motif, as they get additional input from other upstream neurons (US) that are not directly linked to sensory neurons (gray circles; between 17% and 37%), as well as from MB KCs (purple circles; between 36% and 49%) and little MBON feedback (dark-red circle; between 1% and 2%). Examination of the different input types of DL1 DANs presented here shows that DAN-f1 and DAN-g1 receive the highest proportion of sensory input relative to other DANs. Even by looking at the sensory-to-DAN interneurons individually, one can again see the familiar pattern of differences and partial overlap (*Figure 7D*). DAN-d1, DAN-f1, und DAN-g1 all get input from interneuron pair 1 in contrast to DAN-c1. However again, DAN-f1 and DAN-g1 are most strongly connected to this neuronal pair (indicated by the thickness of the lines). DAN-f1 and DAN-g1 are, therefore, also the most similar in terms of their gustatory input (due to their connection via interneuron pair 1), despite all the clearly recognizable differences.

For the pPAM DANs, we found the same general connectivity principles as for DL1 DANs regarding input from MBONs, KCs, and non-sensory upstream neurons (*Figure 7B*). Please note that the DAN-h1 neuron is not present at the early L1 EM brain volume and thus cannot be included in this evaluation (*Figure 7B and E*). In contrast, however, it is striking that pPAM DANs seem to get input from more interneurons connected to the sensory system (up to 16 for DAN-j1) signaling additional internal somatosensory and $CO_2$ information. A closer inspection of the sensory-to-DAN interneurons showed that 27 of the 35 neurons are specifically innervating only DL1 or pPAM DANs (*Figure 7C*). Only 8 out of 35 interneurons connect to both clusters. Based on a calculated hub score, sensory-to-DAN interneurons pair 1 (FB2IN-12) and pair 5 (FFN-21) were likely most significant for instructing DL1 DANs with sensory information. Whereas pairs 10 (FFN-17) and 11 (FFN-12) were – based on connectivity – most important in instructing DL1 and pPAM DANs. For the three pPAM DANs sensory-to-DAN interneuron pairs 16, 17, 18, 19, and 20 resulted in the highest hub scores (FFN-37, FFN-4, FFN-5, FFN-18, and FB-4).

The difference in cellular connectivity of DL1 and pPAM DANs is also reflected in different local input brain hubs: axo-dendritic synapses of sensory-to-DAN interneurons with DL1 DANs formed a characteristic hook shape in the superior protocerebrum; in contrast, pPAM DANs get synaptic input from sensory-to-DAN interneurons in the anterior inferior protocerebrum (*Figure 7F*).

## Discussion
### A functional assignment for individual DANs of the DL1 cluster in their ability to encode a salt punishment teaching signal

The discovery of DL1 neurons as aversive teaching signal mediators in larval *Drosophila* (*Eschbach et al., 2020*; *Schroll et al., 2006*; *Selcho et al., 2009*; *Weiglein et al., 2021*) complements previous findings that demonstrate the sufficiency of four DANs from the pPAM cluster as an appetitive

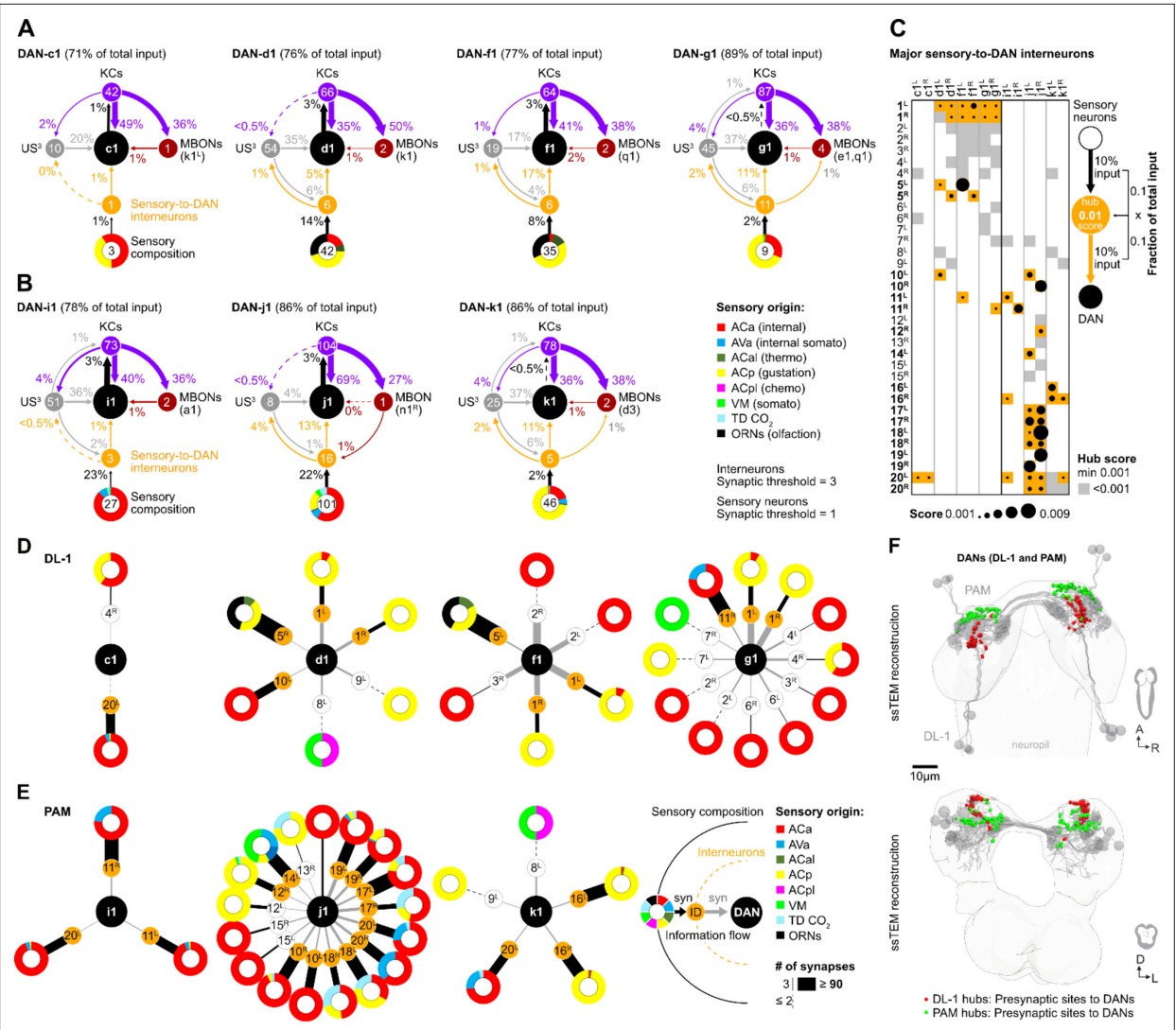

**Figure 7.** Interneurons and hub analysis of sensory to dopaminergic neuron (DAN) pathways. (**A, B**) Schematic graph representation of individual DL1 (**A**) and pPAM (**B**) DANs. The outer ring at the bottom of each scheme represents the sensory composition of neurons targeting sensory-to-DANs interneurons (orange circle). The type of sensory information is encoded by the respective color (ACa: anterior central sensory compartment, AVa: ventral anterior sensory compartment, ACal: lateral anterior central sensory compartment, ACp: posterior anterior central sensory compartment, ACpl: posterior-lateral anterior central sensory compartment, VM: ventromedial sensory compartment, TD $CO_2$: tracheal dendritic neurons responding to $CO_2$, ORNs: olfactory receptor neurons). DAN input neurons are shown to get no direct sensory input (gray circles). Individual DANs are shown in the middle of the scheme as black circles. They are connected to mushroom body (MB) KC (purple circles), which in turn connect to mushroom body output neuron (MBONs; dark red circles). Arrows indicate the direction of the synaptic connection and its strength (coded by arrow thickness). Numbers in circles indicate number of neurons. The percentages indicate the proportion relative to the total input that the cell receives from the specific neuronal partners. For example, DAN-f1 receives 17% of its input from six different sensory-to-DAN interneurons (yellow circle), 17% from 19 other, non-sensory interneurons, 41% from 64 MB KCs, and 2% from two MBONs. (**C**) Dot plot showing the importance of interneurons acting as sensory to DAN hub. Dot size was calculated using the fraction of total input an interneuron receives from sensory neurons multiplied by the fraction of total input this interneuron gives to a DAN. Colored backgrounds of dots are highlighted in orange for the connections with a hub size of 0.001 or above. (**D, E**) Schematic of graph representation. The outer ring represents the sensory composition of neurons targeting upstream neurons of DANs. The type of sensory information is encoded by the respective color. Synaptic threshold for upstream interneurons of DANs = 3 and of upstream sensory neurons = 1. Line thickness to interneurons and targets represents the percentage of synaptic input. White or orange circles connected to the outer ring represent the interneuron layer. The inner ring represents individual target neurons, DL1 and pPAM DANs. The identity of each DAN and interneuron is given by the label in its related circle. (**F**) EM reconstruction of DL1, and pPAM DANs (gray) highlighting their presynaptic sites in red (for DL1 DANs) and green (for pPAM DANs). At the top, a horizontal view of the brain is shown. At the bottom, a frontal view of the brain is shown.

The online version of this article includes the following figure supplement(s) for figure 7:

**Figure supplement 1.** A sensory neuron to DAN map based on EM connectivity.

**Table 1.** A summary of the characteristics of the individual DL1 dopaminergic neurons (DANs).

| | | DAN-c1 | DAN-d1 | DAN-f1 | DAN-g1 |
|---|---|---|---|---|---|
| Behavior | DAN ablation | No | No | No | No |
| | DAN substitution | No | No | Yes | Yes |
| | DAN inhibition | No | No | No | Yes |
| Physiology | | NaCl ↑ Fructose ↑ | NaCl ↑ | Fructose ↓ | NaCl ↑ |
| Connectomics | | → Increasing # of sensory-to-DAN interneurons → | | | |

reinforcement signal in these animals (*Rohwedder et al., 2016*). This dichotomy demonstrates an organizational principle shared by adult *Drosophila* and larvae (*Aso et al., 2012*; *Aso and Rubin, 2016*; *Aso et al., 2010*; *Burke et al., 2012*; *Claridge-Chang et al., 2009*; *Das et al., 2014*; *Huetteroth et al., 2015*; *Liu et al., 2012*; *Schleyer et al., 2020*; *Schroll et al., 2006*; *Selcho et al., 2009*; *Weiglein et al., 2021*; *Yamagata et al., 2015*), despite a significant reduction in cell numbers. The principle appears to apply to mammals as well (*Groessl et al., 2018*; *Lammel et al., 2012*; *Menegas et al., 2018*; *Schultz, 2015*). But how do individual DL1 neurons contribute to the overall teaching signal? Is the signal evenly distributed across multiple cells, supporting the mass action hypothesis, or do individual DANs specifically encode for the salt stimulus (*Rohwedder et al., 2016*)? Our results show that the four DL1 DANs perform this function in a mixture of both forms. There is no unique labeled line because both ablation of each individual DAN (*Figure 2—figure supplement 1D–G*, *Figure 4—figure supplement 1*) and the acute optogenetic inhibition (*Figure 6*) do not abolish the odor-high salt memory. The teaching signal is encoded in a redundant manner within the DL1 cluster, yet its distribution among the four DANs is non-uniform and relies heavily on the functionalities of DAN-g1 and DAN-f1 (*Table 1*, *Figures 4–6*). Each DAN possesses a distinct identity, which is defined by its specific, albeit partially overlapping, neuronal input circuitry and physiological response to salt (*Table 1*, *Figures 3 and 7*).

DAN-c1 responds to both positive and negative gustatory stimuli (*Figure 3A*). However, the activity of the cell is not sufficient or necessary for aversive memory (*Figure 4—figure supplement 1A and E*, *Figures 5A and 6A*). These results suggest that DAN-c1 activity does not encode a taste-specific teaching signal, which is supported by EM analysis, as this particular cell is only weakly connected to three sensory neurons via a single sensory-to-DAN interneuron (*Figure 7A and D*). However, this interpretation might depend on the type and duration of DAN stimulation, as it was recently shown that - as in our case - optogenetic activation does not induce aversive memory, but thermogenetic 30 min stimulation in a non-reciprocal one trial paradigm leads to aversive memory (*Qi et al., 2024*). In addition, it could also be that the function of DAN-c1 is more to integrate state-dependent network changes, similar to the role of some DANs in innate and state-dependent preference behavior of adults (*Boto et al., 2019*; *Cazalé-Debat et al., 2024*; *Cohn et al., 2015*; *Lewis et al., 2015*; *Mohammad et al., 2024*; *Siju et al., 2020*). DAN-d1 activity appears to have a negligible impact on the formation of odor-high salt memories in the paradigms we tested. Even though the neuron responds to high salt concentrations (*Figure 3B*) and receives input from 42 sensory neurons via six sensory-to-DAN interneurons (*Figure 7A and D*), its activity did not induce an aversive memory (*Figure 5B*) and was not found to be essential for odor-high salt memory (*Figure 2—figure supplement 1D and E*, *Figure 4—figure supplement 1B and F*, *Figure 5B*). Nonetheless, it is possible that DAN-d1 could instruct other salt-dependent memory types, established after non-associative training, second-order conditioning, or reversal learning (*Kacsoh et al., 2015*; *König et al., 2019*; *Mancini et al., 2019*; *Paranjpe et al., 2012*; *Tabone and de Belle, 2011*; *Yamada et al., 2023*). Other learning experiments in larvae (*Eschbach et al., 2020*; *Weiglein et al., 2021*) demonstrate that DAN-d1 activation can indeed guide the formation of aversive odor memory. However, these studies employ three training repetitions rather than a single repetition, as in our approach. Additionally, they utilize absolute conditioning with a single odor, in contrast to our protocol of differential conditioning involving two odors. Furthermore, their testing does not incorporate salt as an aversive gustatory stimulus. The observed differences in results suggest that DAN-d1 activation may only establish aversive memory if the stimulation is applied multiple times and is more distinctly presented during training. In combination, DAN-f1 activity and DAN-g1 seem to have a crucial function in high salt punishment. DAN-f1 is inhibited by

sugar (*Figure 3C*) and obtains input from 35 sensory neurons through six sensory-to-DAN interneurons (*Figure 7A and D*). Despite ablation and acute suppression of activity having no impact on the retention of a memory related to a high salt stimulus (*Figure 2—figure supplement 1C, E and G*, *Figure 4—figure supplement 1C and G*, *Figure 6C*), DAN-f1 activity can instruct the formation of an aversive memory (*Figure 5C*). However, there is the possibility that the impact of the ablation experiments could be more pronounced than what is observed here, given the potential for developmental compensation. This has been anatomically demonstrated in the adult dopamine mutants by serotonergic cells (*Niens et al., 2017*). The identified function for DAN-f1 activity is consistent with previous research, although their experimental designs differ from our own in the test situation (*Eschbach et al., 2020*; *Weiglein et al., 2021*). Consequently, it is plausible that DAN-f1 activity encodes for a teaching signal that carries additional information beyond high salt. It is evident that DAN-g1 exerts the most potent effect. Following exposure to high salt, this cell becomes active (*Figure 3D*), is crucial for the formation of an odor-high salt memory (*Figure 6D*) and can even establish an aversive memory that can be recalled on a high salt-containing test plate (*Figure 5D*). Additionally, it receives extensive sensory input from nine neurons through 11 distinct interneurons (*Figure 7A and D*). Therefore, DAN-g1 is the only cell of the DL1 cluster that satisfies all prerequisites to independently encode the high salt teaching signal. In summary, the DL1 cluster appears to be arranged in a reasonable functional configuration. The instructive signal is encoded by a mere four cells, with the requisite of partial redundancy to preclude complete loss of function in the event of single-cell failure. Nevertheless, the cells manifest variation, thereby covering different and cumulative characteristics of stimulation.

However, these cellular-level interpretations are inevitably oversimplified. The MB054B split line occasionally exhibits a weak indication of a third cell (*Figure 2J*), introducing a degree of variability in the experiments. Similarly, DAN-f1 demonstrates no direct response to salt (*Figure 3C*), suggesting it is not essential for salt learning, yet it triggers a potent aversive memory (*Figure 5C*). Nevertheless, it is worth mentioning that the physiological and EM analyses were conducted using first instar larvae, whereas the light microscopy experiments and behavioral tests were performed with third instar larvae. Hence, it is possible that developmental-specific alterations may have occurred, which were not accounted for in this study. Regrettably, our understanding of how salt is perceived in larvae remains limited (*Liu et al., 2003*). Assuming peripheral organ perception as primary, the left and right interneuron 1 (also called FB2IN-12), connected to DAN-d1, DAN-f1, and DAN-g1, is presumed to be centrally important as it encodes for the main peripheral gustatory information (encoded by yellow color, *Figure 7C*, first and second line). However, these DANs display no response to low salt concentrations and varying responses to high salt concentrations and fructose. In contrast, DAN-c1 is the only cell that responds to high and low concentrations of salt (*Figure 3A*) but is not connected to interneuron pair 1 or other sensory-to-DAN interneurons (except interneuron 20 L; *Figure 7C*). Further experiments are, therefore, necessary to refine our initial cellular-level description and comprehend the functional spectrum of individual DL1 DANs more comprehensively.

Furthermore, other factors must also be taken into account. Blocking almost all TH-positive cells does not result in the complete disappearance of the salt memory. Instead, it is merely reduced by half (*Figure 1H and I*). There could be several reasons for this outcome, or a combination thereof. For example, it is plausible that the UAS-GtACR2 effector does not completely suppress the activity of dopaminergic neurons. Additionally, the memory may encompass different types (*Widmann et al., 2016*), not all of which are associated with dopamine function. It is also noteworthy that TH-Gal4 does not encompass all dopaminergic neurons - even a neuron from the DL1 cluster is missing (*Selcho et al., 2009*). Considering that we are using high salt concentrations in this experiment, it is also conceivable that no taste-driven memories are formed solely due to the systemic effects of salt (e.g. increased osmotic pressure). These effects naturally complicate the interpretation of the data at the single-cell level.

## The functioning of the DL1 cluster DANs as a multimodal node for punishment

Larvae, of course, perceive not only high salt concentrations as punishing, but also other sensory stimuli such as bitter substances (quinines and caffeine), electric shock, temperature, mechanosensory input (vibration), and light (*Aceves-Piña and Quinn, 1979*; *Apostolopoulou et al., 2016*; *Apostolopoulou et al., 2014*; *Eschbach et al., 2011*; *Gerber and Hendel, 2006*; *Khurana et al., 2012*; *Pauls et al.,*

2010; *von Essen et al., 2011*). However, the extent to which the DL1 cluster is involved in coding for most of these aversive teaching signals remains unclear. Currently, comparisons at the level of DANs can only be made for sensory inputs that are dependent on high salt or mechanosensation. Interestingly, similar to high salt concentrations, somatosensory information can also result in the establishment of aversive olfactory memory in larvae (*Eschbach et al., 2020*). The transmission of these somatosensory teaching signals is orchestrated by mechanosensory neurons from the chordotonal organs, class IV multidendritic nociceptive neurons distributed along the body, and multisensory basin neurons (*Hwang et al., 2007*; *Jovanic et al., 2016*; *Ohyama et al., 2015*; *Tracey et al., 2003*). The connectome revealed that these mechanosensory neurons also link to the DL1 cluster through three to four interneurons and elicit the activation of DAN-d1, DAN-f1, and DAN-g1 (*Eschbach et al., 2020*). Upon comparing the interneurons that relay high salt and mechanosensory information to the DL1 cluster, six pairs of interneurons are of critical significance, respectively. These pairs include FFN-20, FB2N-12, FB2N-19, FFN-23, FFN-29, and FB2N-15 for mechanosensory information (*Eschbach et al., 2020*), and FB2N-12, FB2N-18, FB2IN-11, FB2IN-6, FFN-21, and FFN-24 for high salt concentrations (*Figure 7*; *Figure 7—figure supplement 1A*, rightmost column). Interestingly, only one pair of interneurons - FB2N12 - is shared between the two types of information and the other five types are specific to the sensory modality. The wiring patterns of DAN input neurons exhibit a notable consistency, with a limited number of neurons dedicated to a specific sensory modality, and single neurons shared across different modalities. This results in a unique combination of cellular codes that are partly redundant and partly specific for the four DL1 DANs. While certain DANs may hold greater significance for a particular information, such as DAN-d1 for somatosensory information, the DAN-g1 cell appears to play a general central role for encoding an aversive teaching signal. DAN-g1 responds to all aversive stimuli tested thus far and its activity has proven informative for all aversive memories analyzed to date. In order to gain a deeper comprehension of the processing of diverse sensory modalities in the larval brain, it is crucial to broaden the analytical framework presented in our work to encompass visual stimuli, various bitter substances, and temperature stimuli.

However, the overall depiction presented above of the information flow and processing of DL1 DANs is an oversimplification. Indeed, DL1 and pPAM DANs are among the most complex and highly recurrent neurons in the brain (*Winding et al., 2023*). This exceptional level of recurrent connectivity enables DANs to provide high-dimensional feedback, which enables them to encode a diverse range of features and engage in parallel computations. Such computations can guide a working memory, contain feedback from neurons that integrate both learned and innate values, and receive long-range feedback from descending neurons that encode motor commands; all in addition to polysynaptic feedforward inputs from the entire set of sensory modalities.

## A cellular comparison of DAN function across *Drosophila* metamorphosis

A recent study conducted by Truman and colleagues explored the integration, or lack thereof, of individual input and output neurons from the larval MB into the adult network following metamorphosis (*Truman et al., 2023*). The study demonstrated that while all four DANs of the pPAM cluster perished, three out of the four cells from the DL1 cluster persisted as part of the adult MB circuitry. Only DAN-f1 altered its innervation in the adult brain, relocating from the larval MB to the adult superior medial protocerebrum. In contrast, DAN-c1, DAN-d1, and DAN-g1 maintained their function as MB DANs in the adult system, identified as PPL1-γ1pedc, PPL1-γ2α'1, and PPL1-γ1, respectively. Due to the lack of appropriate genetic tools, data on the function of PPL1-γ1 is scarce (*Aso et al., 2014*). Therefore, a functional comparison between larval and adult DANs is only possible for the DAN-c1/PPL1-γ1pedc and DAN-d1/PPL1-γ2α'1 combination. Embryonic-born DAN-c1/PPL1-γ1pedc, DAN-d1/PPL1-γ2α'1, and DAN-g1/PPL1-γ1 exhibit sustained activity and respond to various sensory stimuli, including electric shock, temperature, or bitter taste, that can instruct short-lasting aversive memories (*Aso et al., 2012*; *Aso and Rubin, 2016*; *Aso et al., 2010*; *Claridge-Chang et al., 2009*; *Das et al., 2014*; *Galili et al., 2014*; *Huang et al., 2024*; *Kirkhart and Scott, 2015*; *Tomchik, 2013*; *Villar et al., 2022*; *Vrontou et al., 2021*). Despite the massive reorganization of the mushroom body, the basic function of DAN-c1/PPL1-γ1pedc and DAN-d1/PPL1-γ2α'1, mediating punishment teaching signals that instruct short-term memories, remains the same. Evidence indicates that the underlying molecular signaling pathways maintain their identity and kinetics throughout development, with larval

and adult interstimulus interval curves for DAN-d1/PPL1-γ2α'1 being almost identical (*Aso and Rubin, 2016*; *Huang et al., 2024*; *Weiglein et al., 2021*). During adulthood, PPL1-DANs are responsible for numerous other functions, including innate olfactory responses, suppression of appetite memory in response to nutritional status, pre-exposure learning, memory reconsolidation, and forgetting (*Berry et al., 2015*; *Berry et al., 2018*; *Felsenberg et al., 2017*; *Jacob et al., 2021*; *Krashes et al., 2009*; *McCurdy et al., 2021*; *Plaçais et al., 2012*; *Siju et al., 2020*; *Tian et al., 2016*; *Vrontou et al., 2021*). Additionally, they are involved in various behaviors like locomotion and sleep (*Berry et al., 2015*).

An open question is whether larval DL1 DANs encode these same functions to a similar extent. Our study establishes a fundamental framework that can be leveraged to investigate this issue. Considering the numerical simplicity of the larval nervous system, the available genetic tools, and the complete connectome, it may be possible to attain an understanding of the larval dopaminergic system at the level of single cells and single synapses. It is plausible that nature has a limited number of effective neural circuit solutions for complex cognitive and behavioral problems, such as the remarkable ability of brains to learn from a small number of examples. Therefore, comprehending the larval dopaminergic system may provide insight into the shared circuit motifs that exist across the animal kingdom.

# Materials and methods

**Key resources table**

| Reagent type (species) or resource | Designation | Source or reference | Identifiers | Additional information |
|---|---|---|---|---|
| Genetic reagent (*D. melanogaster*) | $w^{1118}$ | N/A | RRID:BDSC:3605 | control genotype |
| Genetic reagent (*D. melanogaster*) | UAS-mCD8::GFP | *Pfeiffer et al., 2010* | RRID:BDSC:32194 | effector line |
| Genetic reagent (*D. melanogaster*) | UAS-mCD8::GFP; nSyb-LexA, IOP-mRFP/TM6B | combined from *Burke et al., 2012* and *Riabinina et al., 2015* | N/A | effector line |
| Genetic reagent (*D. melanogaster*) | UAS-mCD8::GFP; mb247-LexA, IOP-mRFP/TM3,Sb | *Burke et al., 2012* | N/A | effector line |
| Genetic reagent (*D. melanogaster*) | UAS-hid,rpr | *Abbott and Lengyel, 1991*; *White et al., 1996* | N/A | effector line |
| Genetic reagent (*D. melanogaster*) | UAS-GtACR2 | *Mohammad et al., 2017* | RRID:BDSC:92984 | effector line |
| Genetic reagent (*D. melanogaster*) | UAS-ChR2$^{XXL}$ | *Dawydow et al., 2014* | RRID:BDSC:58374 | effector line |
| Genetic reagent (*D. melanogaster*) | UAS-GCaMP6m | *Chen et al., 2013* | RRID:BDSC:42748 | effector line |
| Genetic reagent (*D. melanogaster*) | TH-Gal4 | *Friggi-Grelin et al., 2003* | RRID:BDSC:95268 | driver line |
| Genetic reagent (*D. melanogaster*) | R58E02 | *Jenett et al., 2012* | RRID:BDSC:41347 | driver line |
| Genetic reagent (*D. melanogaster*) | MB054B | *Eschbach et al., 2020* | N/A | split-line |
| Genetic reagent (*D. melanogaster*) | MB065B | *Eschbach et al., 2020* | RRID:BDSC:68281 | split-line |
| Genetic reagent (*D. melanogaster*) | MB143B | *Eschbach et al., 2020* | N/A | split-line |
| Genetic reagent (*D. melanogaster*) | MB145B | *Eschbach et al., 2020* | N/A | split-line |
| Genetic reagent (*D. melanogaster*) | MB328B | *Eschbach et al., 2020* | N/A | split-line |
| Genetic reagent (*D. melanogaster*) | SS01702 | *Eschbach et al., 2020* | N/A | split-line |

*Continued on next page*

*Continued*

| Reagent type (species) or resource | Designation | Source or reference | Identifiers | Additional information |
|---|---|---|---|---|
| Genetic reagent (*D. melanogaster*) | SS01716 | *Eschbach et al., 2020* | N/A | split-line |
| Genetic reagent (*D. melanogaster*) | SS01958 | *Eschbach et al., 2020* | N/A | split-line |
| Genetic reagent (*D. melanogaster*) | SS02160 | *Eschbach et al., 2020* | N/A | split-line |
| Genetic reagent (*D. melanogaster*) | SS02180 | *Eschbach et al., 2020* | N/A | split-line |
| Genetic reagent (*D. melanogaster*) | SS21716 | *Eschbach et al., 2020* | N/A | split-line |
| Genetic reagent (*D. melanogaster*) | SS24765 | *Eschbach et al., 2020* | N/A | split-line |
| Chemical compound, drug | Phosphate Buffered Saline | Sigma-Aldrich | N/A | Cat. no. P4417 |
| Chemical compound, drug | Triton X-100 | Sigma-Aldrich | CAS: 9002-93-1 | Cat. no. X100 |
| Chemical compound, drug | 4% Formaldehyde | Thermo Scientific | CAS: 50-00-0 | Cat. no. 047392.9M |
| Chemical compound, drug | Normal goat serum | Sigma-Aldrich | N/A | Cat. no. G9023 |
| Chemical compound, drug | Xylene | Sigma-Aldrich | CAS:1330-20-7 | Cat. no. 247642 |
| Chemical compound, drug | dibutyl phthalate in xylene | Sigma-Aldrich | CAS: 84-74-2 | Cat. no. 06522 |
| Chemical compound, drug | NaCl | VWR Chemicals | CAS: 7647-14-5 | Cat. no. 27810.364 |
| Chemical compound, drug | D-Fructose | Sigma-Aldrich | CAS: 57-48-7 | Cat. no. 47740 |
| Chemical compound, drug | Agarose | Sigma-Aldrich | CAS: 9012-36-6 | Cat. no. A9539 |
| Chemical compound, drug | Quinine | Sigma-Aldrich | CAS: 207671-44-1 | Cat. no. Q1250 |
| Chemical compound, drug | Amyl acetate | Sigma-Aldrich | CAS: 628-63-7 | Cat. no. 46022 |
| Chemical compound, drug | Paraffin oil | Sigma-Aldrich | CAS: 8012-95-1 | Cat. no. 76235 |
| Chemical compound, drug | Hexyl acetate | Sigma-Aldrich | CAS: 142-92-7 | Cat. no. 108154 |
| Chemical compound, drug | Benzaldehyde | Sigma-Aldrich | CAS: 100-52-7 | Cat. no. 12010 |
| Chemical compound, drug | all-*trans*-retinal | Sigma-Aldrich | CAS: 116-31-4 | Cat. no. R2500 |
| Antibody | rat anti-N-Cadherin | Hybridoma | RRID:AB_528121 | Cat. no. DN-Ex #8 |
| Antibody | rabbit anti-GFP | Life Technologies | RRID:AB_221570 | Cat. no. A6455 |
| Antibody | mouse 4F3 anti-DLG | Hybridoma | RRID:AB_528203 | Cat. no. 4F3 anti-discs large |
| Antibody | goat anti-rat Alexa Fluor 647 | Life Technologies | RRID:AB_141778 | Cat. no. A21247 |
| Antibody | goat anti-rabbit Alexa Flour 488 | Life Technologies | RRID:AB_143165 | Cat. no. A11008 |
| Antibody | goat anti-mouse Alexa Fluor 568 | Life Technologies | RRID:AB_2535804 | Cat. no. A21235 |
| Antibody | mouse anti-TH | Immunostar | RRID:AB_572268 | Cat. no. 22941 |
| Antibody | goat anti-mouse Cy3 | Life Technologies | RRID:AB_10373848 | Cat. no. A10521 |
| Software, algorithm | Fiji version 1.53 c (64-bit) | NIH | https://fiji.sc/ | |
| Software, algorithm | GraphPad Prism 8.4.3 | GraphPad Software, La Jolla, CA | https://www.graphpad.com/scientific-software/prism/ | |

*Continued on next page*

Continued

| Reagent type (species) or resource | Designation | Source or reference | Identifiers | Additional information |
|---|---|---|---|---|
| Software, algorithm | Adobe Photoshop | Adobe Systems, San Jose, CA | https://www.adobe.com/de/products/photoshop.html | |
| Software, algorithm | Affinity Publisher 2.1.1 | Serif (Europe) Ltd., Nottingham, UK | https://affinity.serif.com/de/publisher/ | |
| Software, algorithm | ZEN 2.3 software | Carl Zeiss Microsocopy Germany GmbH | https://www.zeiss.de/mikroskopie/produkte/mikroskopsoftware/zen-lite/zen-lite-download.html | |

## Fly strains

Flies were raised and maintained on *Drosophila* standard food at 25 °C, 60–80% relative humidity, and a 14/7 hr light/dark cycle. For anatomical analysis, UAS-mCD8::GFP (Bloomington stock center no. 32194) (*Selcho et al., 2009*), UAS-mCD8::GFP;mb247-LexA,lexAop-mRFP/TM3,Sb (*Burke et al., 2012*) and UAS-mCD8::GFP;nSyb-LexA,lexAop-mRFP/TM6B (*Burke et al., 2012*; *Riabinina et al., 2015*) were used to analyze the morphology of DANs. For behavioral experiments, UAS-hid,rpr (*Abbott and Lengyel, 1991*; *White et al., 1996*), UAS-GtACR2 (*Mohammad et al., 2017*) (Bloomington stock center no. 92984), and UAS-ChR2$^{XXL}$ (*Dawydow et al., 2014*) (Bloomington stock center no. 58374) were used to ablate or optogenetically silence or activate neurons by blue light (470 nm). UAS-GCaMP6m (*Chen et al., 2013*) (Bloomington stock center no. 42748) flies were used for Ca$^{2+}$-imaging experiments. The Gal4 strains TH-Gal4 (*Friggi-Grelin et al., 2003*) and R58E02-Gal4 (*Rohwedder et al., 2016*) (Bloomington stock center no. 41347) were used to manipulate different sets of DANs. Split-Gal4 lines MB054B, MB065B, MB328B, MB143B, MB145B, SS02160, SS02180, SS01716, SS01958, SS21716, SS24765, SS01702 (*Eschbach et al., 2020*; *Pfeiffer et al., 2010*; *Saumweber et al., 2018*) were used for anatomical, physiological, and behavioral experiments. $w^{1118}$ flies (Bloomington stock center no. 3605) were used to obtain heterozygous driver and effector control groups. See key resources table for more information.

## Anatomical analysis

### Immunostaining

To confirm expression patterns of R58E02-Gal4, TH-Gal4, and MB054B split-Gal4, flies were crossed to UAS-mCD8::GFP. To assess the functionality of the ablation by effector UAS-hid,rpr, split-Gal4 MB054B was crossed with the latter. Furthermore, MB054B>+ and +>UAS-hid,rpr were anatomically examined for validation. All flies were raised at 25 °C for 7 d before dissection. Third instar larvae (wandering stage) were put on ice and dissected in PBS (Phosphate Buffered Saline; Sigma Aldrich, cat. no. P4417). Brains were fixed in 4% formaldehyde solution (in PBS, Thermo Scientific, cat. no. 047392.9M) for 20 min at room temperature. After eight rinses in PBT (PBS with 3% Triton X-100, Sigma Aldrich, cat. no. X100), brains were blocked with 5% normal goat serum (NGS, Sigma Aldrich, cat. no. G9023) in PBT for 1 hr at room temperature and incubated for 48 hr with primary antibodies at 4 °C. Before application of the secondary antibodies for at least 24 hr at 4 °C, brains were washed seven times with PBT. After that, larval brains were again washed eight times with PBT and mounted on poly-L-lysin-coated coverslips (Janelia FlyLight DPX mounting receipt: https://www.janelia.org/project-team/flylight/protocols), dehydrated through a series of increasing concentrations of ethanol and cleared three times for 5 min in xylene (Sigma Aldrich, cat. no. 247642). At the end, larval brains were mounted in DPX mounting medium (dibutyl phthalate in xylene, Sigma Aldrich, cat. no. 06522) and left to rest in the dark for at least 24 hr before imaging. Primary antibodies were: rat anti-N-Cadherin (1:50; Hybridoma, cat. no. DN-Ex #8), rabbit anti-GFP (1:1000; Life Technologies, cat. no. A6455), mouse 4F3 anti-DLG (1:200; Hybridoma, cat. no. 4F3 anti-discs large) and mouse anti-TH (1:50, Immunostar, cat. no. 22941). Secondary antibodies were: goat anti-rat Alexa Fluor 647 (1:500 or 1:250, for anti-N-Cadherin, Life Technologies, cat. no. A21247), goat anti-rabbit Alexa Fluor 488 (1:500, for anti-GFP, Life Technologies, cat. no. A11008), goat anti-mouse Alexa Fluor 568 (1:500, for anti-DLG, Life Technologies, cat. no. A10037), and goat anti-mouse Cy3 (1:200, for anti-TH, Life Technologies, cat. no. A10521).

## Native fluorescence

For native fluorescence, we used a previously described protocol for adult flies (*Pitman et al., 2011*) and adapted it to larvae. All split-Gal4 lines were crossed to UAS-mCD8::GFP;mb247-LexA,lexAop-mRFP/TM3,Sb, while split-Gal4 line MB054B was additionally crossed to UAS-mCD8::GFP;nSyb-LexA,lexAop-mRFP/TM6B and raised at 25 °C for 7 d. Third instar larvae (wandering stage) were put on ice and dissected in PBS. Brains were fixed under vacuum in 4% formaldehyde solution (in PBS) for 25 min at room temperature. After five washes in PBT (PBS with 3% Triton X-100), larval brains were rinsed twice directly and once for 10 min under vacuum in PBS at room temperature. Samples were mounted on poly-L-lysin-coated coverslips and embedded in Vectashield (Vector Laboratories, cat. no. H-1000–10). Slides were stored in the dark and cold for at least 24 hr before imaging.

## Confocal microscopy

Confocal microscopy was conducted on a Zeiss LSM800 confocal laser scanning microscope with ZEN 2.3 software. All images were projected and adjusted with ImageJ (Fiji is just ImageJ, Version 1.53 c, Java 1.8.0_172 (64-bit)). Final Figures were arranged and labeled in Affinity Publisher 2.1.1.

## Functional imaging and microfluidics

For larval imaging experiments, adult flies were transferred to larvae collection cages (Genesee Scientific) containing grape juice agar plates and 180 mg of fresh yeast paste per cage. Flies were allowed to lay eggs on the agar plate for 1–2 d before the plate was removed for collection of larvae. Calcium imaging experiments were performed in L1 larvae (2 d AEL). We used a previously described method for microfluidic delivery of chemicals in aqueous form with simultaneous imaging of calcium activity in intact larvae (*Si et al., 2019*). All experiments used an 8-channel microfluidic chip equipped with a vacuum port to stabilize the animal's head. The same tastants were used as in the behavioral experiments (NaCl: VWR Chemicals, cat. no. 27810.364; D-Fructose: Sigma Aldrich cat. no. 47740). Stimuli consisted of 5–10 s pulses interleaved with 15 s water washout periods. An L1 larva was washed with deionized water and loaded into the microfluidic device using a 1 mL syringe filled with Triton X-100 (0.1% [v/v]) solution. The animal was pushed to the end of the loading channel with its dorsal side facing the objective. GCaMP6m signal was recorded using an inverted Nikon Ti-E spinning disk confocal microscope and a 60X water immersion objective (NA 1.2). A CCD microscope camera (Andor iXon EMCCD) captured frames at 30 Hz. Recordings from at least 5–7 larvae were collected for each condition. For each larva, the average response over 5 s around the peak value during stimulus presentation was compared to chance level analyzed via normal t-test. Responses during stimulation were normalized based on the baseline response 10 s before the stimulation. p-values for each experiment are given in the respective figure panel. Each graph shows the mean calcium signal plotted as the relative response strength ΔF/F and the related standard error of the mean on the y-axis. The time in seconds is given below each graph on the x-axis. The gray box indicates the duration of the stimulus application. The sample size of each group (N=5–7) is given above each row. n.s. p>0.05; *p<0.05.

## Behavioral experiments

### Associative olfactory learning

Standard experiments were done as described before (*Apostolopoulou et al., 2013*; *Gerber et al., 2013*; *Hendel et al., 2005*; *Michels et al., 2017*; *Widmann et al., 2018*). Learning experiments were conducted on assay plates (85 mm diameter, Sarstedt, cat. no. 82.1472) filled with a thin layer of either 2.5% (w/v) pure agarose solution (Sigma Aldrich, cat. no. A9539) or 2.5% (w/v) agarose plus either 1.5 M sodium chloride solution (VWR Chemicals, cat. no. 27810.364), 2 M D-Fructose solution (Sigma Aldrich cat. no. 47740), or 10 mM quinine solution (quinine hemisulfate; Sigma Aldrich cat. no. Q1250). Before closing and labeling the lids, solutions were let to cool down at room temperature to avoid condensation. Plates were stored at 18 °C and used within 5 d. As olfactory stimuli, we used two different odor combinations: either amyl acetate (AM, Sigma Aldrich, cat. no. 46022) diluted 1:250 in paraffin oil (Sigma Aldrich, cat. no. 76235) and undiluted benzaldehyde (BA, Sigma Aldrich, cat. no. 12010) or benzaldehyde and hexyl acetate (HA, Sigma Aldrich, cat. no. 108154) both diluted 1:100 in paraffin oil. For both odor combinations, 10 µL odor were loaded into custom-made Teflon containers (4.5 mm diameter) with perforated lids (*Scherer et al., 2003*).

A minimum of 30 early third instar larvae were collected and exposed to a first odor (amyl acetate or hexyl acetate) while crawling on pure agarose for 5 min, followed by 5 min exposition to a second odor (benzaldehyde) on agarose medium with either 1.5 M sodium chloride, 10 mM quinine as negative reinforcer or 2 M D-Fructose as positive reinforcer (Odor1/Odor2+). For three cycle training, two more repetitions of training trials were performed. A second group of larvae received reciprocal training (Odor1+/Odor2). Please note that occurrence of odor and reinforcer during training have been randomized to exclude sequential or positional effects. After one or three training cycles, larvae were transferred onto test plates containing either pure agarose (reward learning) or agarose plus salt or quinine (punishment learning) on which odor 1 and odor 2 were represented on opposite sides. After 5 min, larvae were counted on each side of the test plate (#Odor1, #Odor2) or in a 10 mm neutral zone in the middle of the plate (#Neutral). A Preference index (PREF) for each group of larvae was calculated as follows:

$$PREF_{Odor1/Odor2+} = \frac{\#Odor2 - \#Odor1}{\#Total}$$

$$PREF_{Odor1+/Odor2} = \frac{\#Odor1 - \#Odor2}{\#Total}$$

To measure specifically the effect of associative learning, we then calculated the associative performance index (PI) as follows:

$$PI = \frac{PREF_{Odor1/Odor2+} + PREF_{Odor1+/Odor2}}{2}$$

Negative Performance Indices represent aversive associated learning, whereas positive values represent appetitive learning. Division by 2 ensures scores are bound within (–1; 1).

## Odor preference test

Pure agarose (2.5% w/v) plates were cast as previously described (*Apostolopoulou et al., 2013*; *Gerber et al., 2013*; *Michels et al., 2017*; *Scherer et al., 2003*; *Widmann et al., 2018*). Furthermore, the odors were prepared according to the dilutions used in the learning experiments. An odor container (either AM 1:250, BA undiluted, HA 1:100, or BA 1:100) was placed on one side of the agarose plate. An empty container was placed on the opposite side to exclude visual or other side effects. Approximately 30 third instar feeding stage larvae were collected and placed on the plate for 5 min. After this time, larvae were counted, subdivided into larvae on the odor side (#Odor), larvae on the empty container side (#Empty), and larvae in a 10 mm neutral zone in the middle of the plate, as well as larvae that were on the lid of the Petri dish (#Neutral). Preferences were calculated as follows:

$$PREF_{Odor/Empty} = \frac{\#Odor - \#Empty}{\#Total}$$

Positive values indicate a preference for the respective odor, while negative values indicate odor avoidance. The preference tests were performed in different orientations (odor container on plate up/down/left/right) to exclude directional bias.

## Gustatory preference test

High salt-dependent choice behavior experiments were performed using standard methods (*Hendel et al., 2005*; *Huser et al., 2017*; *Huser et al., 2012*; *Niewalda et al., 2008*; *Selcho et al., 2009*; *Widmann et al., 2016*). A 2.5% (w/v) agarose solution was boiled as previously described and a thick layer was poured into Petri dishes. After cooling, half of the agarose in the Petri dish was removed and filled by 2.5% (w/v) agarose solution with 1.5 M sodium chloride. For the choice experiment, about 30 third instar feeding stage larvae were put in the middle of the Petri dish and left them crawl for 5 min. Larvae were then counted as being located on the sodium chloride side (#NaCl), pure agarose side (#Agarose), or a neutral area of about 10 mm width in the middle of the plate (#Neutral), as well as larvae that were on the lid of the Petri dish (#Neutral). The gustatory preference indices for sodium chloride were calculated as follows:

$$\text{PREF}_{\text{NaCl}} = \frac{\#\text{NaCl} - \#\text{Agarose}}{\#\text{Total}}$$

Negatives values for preference indices indicate an aversion to sodium chloride. The gustatory preference tests were performed in different orientations (NaCl solution on plate up/down/left/right) to exclude directional bias.

## Optogenetic substitution experiment

To substitute an actual salt punishment by remotely activating DANs, we used UAS-ChR2$^{\text{XXL}}$. Effector lines were crossed to $w^{1118}$ to obtain appropriate genetic controls. Larvae were maintained in vials with *Drosophila* standard food on 25 °C and wrapped in aluminum foil to ensure development in constant darkness. A group of 30 feeding-stage third-instar larvae were placed onto plates containing 2.5% agarose and exposed to either AM or BA. During the presentation of the first odor, the larvae were exposed to blue light (470 nm, 220 lux) for 5 min. The second odor BA (undiluted) was presented in darkness for 5 min on a pure agarose plate. As described for odor-high salt learning, training was performed reciprocally and the sequence of training trials was alternated across repetitions of the experiment. Furthermore, a control group, identical to the experimental group, underwent no blue light stimulation during the learning paradigm. Data were then scored as above. Please note, tests were done in the presence of 1.5 M sodium chloride to test for the specific quality of the memory. The activation of blue light itself does not cause any disruptive behavioral side effects, as it does not induce an appetitive or aversive memory on its own.

## Optogenetic inhibition of neuronal activity

To acutely block synaptic output, we used UAS-GtACR2. Flies were maintained on standard food supplemented with 0.5 mM all-*trans*-retinal (ATR, Sigma Aldrich, cat. no. R2500) at 25 °C as described before (*Meloni et al., 2020*). Vials were wrapped in aluminum foil to ensure larval development in constant darkness. Groups of about 30 third instar feeding stage larvae received a reciprocal two-odor training as described before, exposed to blue light (470 nm, 1100 lux) during the entire training phase. The sequence of training trials was alternated across repetitions of the experiment. The test was performed for 5 min in darkness on an agarose plate mixed with 1.5 M sodium chloride. Control experiments were performed with the same genotype but with standard food lacking 0.5 mM all-*trans*-retinal. Please note that the blue light exposure itself does not cause any disruptive behavioral side effects, as it does not impair odor-high salt learning of control animals. Data were then calculated and scored as mentioned above.

## Evaluation of DAN input wiring diagrams

Pie charts of sensory profiles were calculated using the percentage of total synaptic input of interneurons as fraction (thereby ignoring other inputs to show distribution of sensory origins). Percentages then give the percentage of total sensory synaptic input to interneurons. The calculation of the hub score was done in the following way: Fraction of total synaptic input from all sensory neurons to defined interneurons (see IDs) was multiplied by the total fraction of input of the DANs from this interneuron. For anatomical reconstructions and visualizations included here, we made use of earlier published data (*Eichler et al., 2017*; *Eschbach et al., 2020*; *Miroschnikow et al., 2018*; *Winding et al., 2023*).

## Statistical analysis

Behavioral results were analyzed in GraphPad Prism 8.4.3. To test for normal distribution, the Shapiro-Wilk test was applied. Groups that did not violate the assumption of normal distribution, were analyzed with an unpaired t-test (comparison of two groups) or a one-way ANOVA followed by Tukey's post hoc test (comparisons between groups larger than two). For nonparametric statistics, the Mann-Whitney test (comparison between two groups) or Kruskal-Wallis followed by Dunn's multiple comparisons tests were applied. Furthermore, one sample t-test for parametric or Wilcoxon signed-rank test for nonparametric data sets were performed to compare means or medians against chance level. Results are visualized in box plots, indicating the median as middle line, 25%/75% quantiles as box boundaries and minimum/maximum performance indices as whiskers. Each data point is represented as

black dot and sample sizes are noted within graphics. Asterisks and 'n.s.' indicate $p<0.05$ and $p>0.05$, respectively. The source data and results of all statistical tests are documented in the source data files.

## Acknowledgements

This work was supported by the Deutsche Forschungsgemeinschaft (Grant No. 441181781, 426722269, 432195391) and by EU funds from the ESF Plus Program (Grant No. 100649752) all to AST. KV was supported by a DFG postdoc grant (Grant No. 345729665). We thank Aravi Samuel for continuous support and discussions. We thank Bert Klagges, Tilman Triphan, Dennis Pauls, Mareike Selcho, and Wolf Huetteroth for discussions and comments. Additionally, we thank Juliane Kinnigkeit for fly care and maintenance.

## Additional information

### Funding

| Funder | Grant reference number | Author |
|---|---|---|
| Deutsche Forschungsgemeinschaft | 441181781 | Andreas S Thum |
| Deutsche Forschungsgemeinschaft | 426722269 | Andreas S Thum |
| Deutsche Forschungsgemeinschaft | 432195391 | Andreas S Thum |
| Deutsche Forschungsgemeinschaft | 345729665 | Katrin Vogt |
| European Social Fund Plus | 100649752 | Andreas S Thum |

The funders had no role in study design, data collection and interpretation, or the decision to submit the work for publication.

### Author contributions

Denise Weber, Conceptualization, Supervision, Investigation, Methodology, Writing – original draft, Writing – review and editing; Katrin Vogt, Conceptualization, Investigation, Methodology, Writing – original draft, Writing – review and editing; Anton Miroschnikow, Michael J Pankratz, Conceptualization, Investigation, Writing – original draft; Andreas S Thum, Conceptualization, Supervision, Funding acquisition, Investigation, Methodology, Writing – original draft, Project administration, Writing – review and editing

### Author ORCIDs

Denise Weber ⓘ https://orcid.org/0000-0002-7341-6280
Katrin Vogt ⓘ https://orcid.org/0000-0002-8763-342X
Anton Miroschnikow ⓘ https://orcid.org/0000-0002-2276-3434
Michael J Pankratz ⓘ https://orcid.org/0000-0001-5458-6471
Andreas S Thum ⓘ https://orcid.org/0000-0002-3830-6596

Reviewer #1 (Public review): https://doi.org/10.7554/eLife.91387.4.sa1
Reviewer #2 (Public review): https://doi.org/10.7554/eLife.91387.4.sa2
Reviewer #3 (Public review): https://doi.org/10.7554/eLife.91387.4.sa3
Author response https://doi.org/10.7554/eLife.91387.4.sa4

## Additional files

### Supplementary files

MDAR checklist

## Data availability

All data generated or analysed during this study are included in the manuscript and supporting files.

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
