## [Editor Report · eLife Assessment]

This comprehensive study presents **important** findings that delineate how specific dopaminergic neurons (DANs) instruct aversive learning in *Drosophila larvae* exposed to high salt through an integration of behavioral experiments, imaging, and connectomic analysis. The work reveals how a numerically minimal circuit achieves remarkable functional complexity, with redundancies and synergies within the DL1 cluster that challenge our understanding of how few neurons generate learning behaviors. By establishing a framework for sensory-driven learning pathways, the study makes a **compelling** and substantial contribution to understanding associative conditioning while demonstrating conservation of learning mechanisms across *Drosophila* developmental stages.

---

## [Referee Report · Reviewer #1 (Public review)]

In this paper Weber et al. investigate the role of 4 dopaminergic neurons of the *Drosophila larva* in mediating the association between an aversive high-salt stimulus and a neutral odor. The 4 DANs belong to the DL1 cluster and innervate non-overlapping compartments of the mushroom body, distinct from those involved in appetitive associative learning. Using specific driver lines for individual neurons, the authors show that activation of the DAN-g1 is sufficient to mimic an aversive memory and it is also necessary to form a high-salt memory of full strength, although optogenetic silencing of this neuron has only a partial phenotype. The authors use calcium imaging to show that the DAN-g1 is not the only DAN responding to salt. DAN-c1 and d1 also respond to salt, but they seem to play no role for the associative memory. DAN-f1, which does not respond to salt, is able to lead to the formation of a memory (if optogenetically activated), but it is not necessary for the salt-odor memory formation in normal conditions. However, when silenced together with DAN-g1, it enhances the memory deficit of DAN-g1. Overall, this work brings evidence of a complex interaction between DL1 DANs in both the encoding of salt signals and their teaching role in associative learning, with none of them being individually necessary and sufficient for both functions.

Overall, the manuscript contributes interesting results that are useful to understand the organization and function of the dopaminergic system. The behavioral role of the specific DANs is accessed using specific driver lines which allow to test their function individually and in pairs. Moreover, the authors perform calcium imaging to test whether DANs are activated by salt, a prerequisite for inducing a negative association to it. Proper genetic controls are carried across the manuscript.

---

## [Referee Report · Reviewer #2 (Public review)]

Summary:

In this work the authors show that dopaminergic neurons (DANs) from the DL1 cluster in *Drosophila larvae* are required for the formation of aversive memories. DL1 DANs complement pPAM cluster neurons which are required for the formation of attractive memories. This shows the compartmentalized network organization of how an insect learning center (the mushroom body) encodes memory by integrating olfactory stimuli with aversive or attractive teaching signals. Interestingly, the authors found that the 4 main dopaminergic DL1 neurons act partially redundant, and that single cell ablation did not result in aversive memory defects. However, ablation or silencing of a specific DL1 subset (DAN-f1,g1) resulted in reduced salt aversion learning, which was specific to salt but no other aversive teaching stimuli tested. Importantly, activation of these DANs using an optogenetic approach was also sufficient to induce aversive learning in the presence of high salt. Together with the functional imaging of salt and fructose responses of the individual DANs and the implemented connectome analysis of sensory (and other) inputs to DL1/pPAM DANs this represents a very comprehensive study linking the structural, functional and behavioral role of DL1 DANs. This provides fundamental insight into the function of a simple yet efficiently organized learning center which displays highly conserved features of integrating teaching signals with other sensory cues via dopaminergic signaling.

Strengths:

This is a very careful, precise and meticulous study identifying the main larval DANs involved in aversive learning using high salt as a teaching signal. This is highly interesting because it allows to define the cellular substrates and pathways of aversive learning down to the single cell level in a system without much redundancy. It therefore sets the basis to conduct even more sophisticated experiments and together with the neat connectome analysis opens the possibility to unravel different sensory processing pathways within the DL1 cluster and integration with the higher order circuit elements (Kenyon cells and MBONs). The authors' claims are well substantiated by the data and balanced, putting their data in the appropriate context. The authors also implemented neat pathway analyses using the larval connectome data to its full advantage, thus providing network pathways that contribute towards explaining the obtained results.

Weaknesses:

Previous comments were fully addressed by the authors.

---

## [Referee Report · Reviewer #3 (Public review)]

The study of Weber et al. provides a thorough investigation of the roles of four individual dopamine neurons for aversive associative learning in the *Drosophila larva*. They focus on the neurons of the DL-1 cluster which already have been shown to signal aversive teaching signals. But the authors go beyond the previous publications and test whether each of these dopamine neurons responds to salt or sugar, is necessary for learning about salt, bitter, or sugar, and is sufficient to induce a memory when optogenetically activated. In addition, previously published connectomic data is used to analyze the synaptic input to each of these dopamine neurons. The authors conclude that the aversive teaching signal induced by salt is distributed across the four DL-1 dopamine neurons, with two of them, DAN-f1 and DAN-g1, being particularly important. Overall, the experiments are well designed and performed, support the authors' conclusions, and deepen our understanding of the dopaminergic punishment system.

Strengths:

(1) This study provides, at least to my knowledge, the first in vivo imaging of larval dopamine neurons in response to tastants. Although the selection of tastants is limited, the results close an important gap in our understanding of the function of these neurons.

(2) The authors performed a large number of experiments to probe for the necessity of each individual dopamine neuron, as well as combinations of neurons, for associative learning. This includes two different training regimen (1 or 3 trials), three different tastants (salt, quinine and fructose) and two different effectors, one ablating the neuron, the other one acutely silencing it. This thorough work is highly commendable, and the results prove that it was worth it. The authors find that only one neuron, DAN-g1, is partially necessary for salt learning when acutely silenced, whereas a combination of two neurons, DAN-f1 and DAN-g1, are necessary for salt learning when either being ablated or silenced.

(3) In addition, the authors probe whether any of the DL-1 neurons is sufficient for inducing an aversive memory. They found this to be the case for two of the neurons, largely confirming previous results obtained by a different learning paradigm, parameters and effector.

(4) This study also takes into account connectomic data to analyze the sensory input that each of the dopamine neurons receives. This analysis provides a welcome addition to previous studies and helps to gain a more complete understanding. The authors find large differences in inputs that each neuron receives, and little overlap in input that the dopamine neurons of the "aversive" DL-1 cluster and the "appetitive" pPAM cluster seem to receive.

(5) Finally, the authors try to link all the gathered information in order to describe an updated working model of how aversive teaching signals are carried by dopamine neurons to the larva's memory center. This includes important comparisons both between two different aversive stimuli (salt and nociception) and between the larval and adult stages.

---

## [Author Response]

The following is the authors’ response to the previous reviews

**Public Reviews:**

**Reviewer #1 (Public review):**
Summary:In this paper Weber et al. investigate the role of 4 dopaminergic neurons of the *Drosophila larva* in mediating the association between an aversive high-salt stimulus and a neutral odor. The 4 DANs belong to the DL1 cluster and innervate non-overlapping compartments of the mushroom body, distinct from those involved in appetitive associative learning. Using specific driver lines for individual neurons, the authors show that activation of the DAN-g1 is sufficient to mimic an aversive memory and it is also necessary to form a high-salt memory of full strength, although optogenetic silencing of this neuron has only a partial phenotype. The authors use calcium imaging to show that the DAN-g1 is not the only DAN responding to salt. DAN-c1 and d1 also respond to salt, but they seem to play no role for the associative memory. DAN-f1, which does not respond to salt, is able to lead to the formation of a memory (if optogenetically activated), but it is not necessary for the salt-odor memory formation in normal conditions. However, when silenced together with DAN-g1, it enhances the memory deficit of DAN-g1. Overall, this work brings evidence of a complex interaction between DL1 DANs in both the encoding of salt signals and their teaching role in associative learning, with none of them being individually necessary and sufficient for both functions.Strengths:Overall, the manuscript contributes interesting results that are useful to understand the organization and function of the dopaminergic system. The behavioral role of the specific DANs is accessed using specific driver lines which allow to test their function individually and in pairs. Moreover, the authors perform calcium imaging to test whether DANs are activated by salt, a prerequisite for inducing a negative association to it. Proper genetic controls are carried across the manuscript.Weaknesses:The authors use two different approaches to silence dopaminergic neurons: optogenetics and induction of apoptosis. The results are not always consistent, but the authors discuss these differences appropriately. In general, the optogenetic approach is more appropriate as developmental compensations are not of major interest for the question investigated.The physiological data would suggest the role of a certain subset of DANs in salt-odor association, but a different partially overlapping set is necessary in behavioral assays (with a partial phenotype). No manipulation completely abolishes the salt-odor association, leaving important open questions on the identity of the neural circuits involved in this behavior.The EM data analysis reveals a non-trivial organization of sensory inputs into DANs, but it is difficult to extrapolate a link to the functional data presented in the paper.

We would like to once again thank Reviewer 1 for the positive assessment of our work and for the valuable suggestions provided on the first revision of the manuscript. In this second revision, we have addressed the linguistic issues and most of the minor comments as recommended. We now hope that the current version of our manuscript meets the reviewer’s expectations both in terms of language and content.

**Reviewer #2 (Public review):**
Summary:In this work the authors show that dopaminergic neurons (DANs) from the DL1 cluster in *Drosophila larvae* are required for the formation of aversive memories. DL1 DANs complement pPAM cluster neurons which are required for the formation of attractive memories. This shows the compartmentalized network organization of how an insect learning center (the mushroom body) encodes memory by integrating olfactory stimuli with aversive or attractive teaching signals. Interestingly, the authors found that the 4 main dopaminergic DL1 neurons act partially redundant, and that single cell ablation did not result in aversive memory defects. However, ablation or silencing of a specific DL1 subset (DAN-f1,g1) resulted in reduced salt aversion learning, which was specific to salt but no other aversive teaching stimuli tested. Importantly, activation of these DANs using an optogenetic approach was also sufficient to induce aversive learning in the presence of high salt. Together with the functional imaging of salt and fructose responses of the individual DANs and the implemented connectome analysis of sensory (and other) inputs to DL1/pPAM DANs this represents a very comprehensive study linking the structural, functional and behavioral role of DL1 DANs. This provides fundamental insight into the function of a simple yet efficiently organized learning center which displays highly conserved features of integrating teaching signals with other sensory cues via dopaminergic signaling.Strengths:This is a very careful, precise and meticulous study identifying the main larval DANs involved in aversive learning using high salt as a teaching signal. This is highly interesting because it allows to define the cellular substrates and pathways of aversive learning down to the single cell level in a system without much redundancy. It therefore sets the basis to conduct even more sophisticated experiments and together with the neat connectome analysis opens the possibility to unravel different sensory processing pathways within the DL1 cluster and integration with the higher order circuit elements (Kenyon cells and MBONs). The authors' claims are well substantiated by the data and balanced, putting their data in the appropriate context. The authors also implemented neat pathway analyses using the larval connectome data to its full advantage, thus providing network pathways that contribute towards explaining the obtained results.Weaknesses:Previous comments were fully addressed by the authors.

We sincerely thank Reviewer 2 for the positive evaluation of our work. We are glad that our responses in the first revision addressed the previous concerns and appreciate the reviewer’s constructive feedback once again.

**Reviewer #3 (Public review):**
Summary:The study of Weber et al. provides a thorough investigation of the roles of four individual dopamine neurons for aversive associative learning in the *Drosophila larva*. They focus on the neurons of the DL-1 cluster which already have been shown to signal aversive teaching signals. But the authors go beyond the previous publications and test whether each of these dopamine neurons responds to salt or sugar, is necessary for learning about salt, bitter, or sugar, and is sufficient to induce a memory when optogenetically activated. In addition, previously published connectomic data is used to analyze the synaptic input to each of these dopamine neurons. The authors conclude that the aversive teaching signal induced by salt is distributed across the four DL-1 dopamine neurons, with two of them, DAN-f1 and DAN-g1, being particularly important. Overall, the experiments are well designed and performed, support the authors' conclusions, and deepen our understanding of the dopaminergic punishment system.Strengths:(1) This study provides, at least to my knowledge, the first in vivo imaging of larval dopamine neurons in response to tastants. Although the selection of tastants is limited, the results close an important gap in our understanding of the function of these neurons.(2) The authors performed a large number of experiments to probe for the necessity of each individual dopamine neuron, as well as combinations of neurons, for associative learning. This includes two different training regimen (1 or 3 trials), three different tastants (salt, quinine and fructose) and two different effectors, one ablating the neuron, the other one acutely silencing it. This thorough work is highly commendable, and the results prove that it was worth it. The authors find that only one neuron, DAN-g1, is partially necessary for salt learning when acutely silenced, whereas a combination of two neurons, DAN-f1 and DAN-g1, are necessary for salt learning when either being ablated or silenced.(3) In addition, the authors probe whether any of the DL-1 neurons is sufficient for inducing an aversive memory. They found this to be the case for two of the neurons, largely confirming previous results obtained by a different learning paradigm, parameters and effector.(4) This study also takes into account connectomic data to analyze the sensory input that each of the dopamine neurons receives. This analysis provides a welcome addition to previous studies and helps to gain a more complete understanding. The authors find large differences in inputs that each neuron receives, and little overlap in input that the dopamine neurons of the "aversive" DL-1 cluster and the "appetitive" pPAM cluster seem to receive.(5) Finally, the authors try to link all the gathered information in order to describe an updated working model of how aversive teaching signals are carried by dopamine neurons to the larva's memory center. This includes important comparisons both between two different aversive stimuli (salt and nociception) and between the larval and adult stages.

We would also like to thank Reviewer 3 for the positive assessment of our work. Many of the constructive comments provided were incorporated into the first revision, contributing significantly to the improved clarity and overall quality of the manuscript.

**Recommendations for the authors:**

**Reviewer #1 (Recommendations for the authors):**
Here are some minor comments (and some semantics that could be addressed to improve the manuscript)Title: is the title correct given that c1 and d1 do not really signal punishment?

We think the title is correct and would like to keep it as it is.

L72 striatum misspelled

We have corrected the error.

L74 constitute instead of provide?

We made the suggested modification in the text.

L129: "But can these four individual DANs also process other sensory modalities?" other then what? What was used before?

We have made the required change, which now allows us to contrast somatosensory and chemosensory information.

L172: (Please refer to the discussion regarding the partial reduction of the memory); would be more natural to explain shortly here, or add a sentence before this parenthesis that point to the effect

We made the requested change in the manuscript and added a short sentence before the parenthesis.

L182: "DL1 neurons convey a dopaminergic aversive teaching signal" you cannot make this statement from just TH-GAL4!

We agree - that's why we have completely revised the sentence and now further restricted it and also refer to further larval and adult published data

L264: "possible redundancy among" I don't think you are testing a redundancy here, it is more likely a developmental compensation.

We made the requested change in the sentence and added a potential developmental compensation as an interpretation of our results.

L296: "to determine if the activation of individual DL1 DANs signals aspects of the natural high salt punishment," - how can the optogenetic activation tell something about aspects of the natural salt punishment? I understand the fact that salt is present, but still I find it inaccurate

Our approach is based on the framework established by Bertram Gerber and colleagues over the past two decades in larval *Drosophila* research. According to this logic, memory recall is dependent on the specific properties of the test context, particularly the type and concentration of the stimulus presented on the test plate. Aversive memory retrieval occurs only when the test conditions closely match those of the training stimulus. Consequently, the larva's behavior on the test plate serves as an indicator of the memory content being recalled. We therefore adhere to this established methodology (Gerber & Hendel, 2006; Schleyer et al., 2011; Schleyer et al., 2015).

L307 "DAN-f1 and DAN-g1 encode aspects of the natural aversive high salt teaching" you cannot conclude that given that f1 does not even respond to salt. I understand the logic of the salt during test, but I think it is still a stretched interpretation

We agree and thus have deleted the sentence.

L310 "Individual DL1 DANs are acutely necessary" this is too general, it seems that only one is

We have changed the title and now clearly state that this is only one DAN of the DL1 cluster.

**Reviewer #2 (Recommendations for the authors):**
In Fig.6 the text flow could be optimized as the authors first mention Fig. 6E,F before they follow up with Fig. 6A-D.

Thanks for bringing this up – we changed it in the revised version of the manuscript. Now 6A-D is mentioned first.

In Fig.6 the finding that optogenetic inactivation but not ablation of DAN-g1 slightly but significantly reduces aversive salt learning suggests that there is an individual contribution of this DAN in this paradigm. The authors emphasize redundancy of DL1 DANs although the effect size seems comparable between DAN-g1 and DAN-f1,g1 silencing.

In response to this concern and the one of reviewer 2, we have revised the section title and removed the final sentence of the section before to avoid placing emphasis on the potential redundancy of DL1 DANs within this results section.

**Reviewer #3 (Recommendations for the authors):**
The authors replied to each issue I raised, and revised their manuscript accordingly. In particular, regarding my major concern (the sufficiency of the neurons for salt-"specific" memories), I think the authors found a good solution.I have no further comments.

We sincerely thank the reviewer for the positive feedback on our revision. We are pleased that the revised manuscript meets the expectations and appreciate the time and effort invested in the review process.